# CD169-mediated restrictive SARS-CoV-2 infection of macrophages induces pro-inflammatory responses

**Sallieu Jalloh**[1☉], **Judith Olejnik**[1,2☉], **Jacob Berrigan**[1], **Annuurun Nisa**[3], **Ellen L. Suder**[1,2], **Hisashi Akiyama**[1], **Maohua Lei**[1], **Sita Ramaswamy**[1], **Sanjay Tyagi**[3], **Yuri Bushkin**[3], **Elke Mühlberger**[1,2‡], **Suryaram Gummuluru**[1‡]*

1 Department of Microbiology, Boston University School of Medicine, Boston, Massachusetts, United States of America, 2 National Emerging Infectious Diseases Laboratories, Boston University, Boston, Massachusetts, United States of America, 3 Public Health Research Institute, New Jersey Medical School, Rutgers University, Newark, New Jersey, United States of America

☉ These authors contributed equally to this work.
‡ EM and SG also contributed equally to this work.
* rgummulu@bu.edu

**Data Availability Statement:** All relevant data are within the manuscript and its Supporting Information files.

## Abstract

Exacerbated and persistent innate immune response marked by pro-inflammatory cytokine expression is thought to be a major driver of chronic COVID-19 pathology. Although macrophages are not the primary target cells of SARS-CoV-2 infection in humans, viral RNA and antigens in activated monocytes and macrophages have been detected in post-mortem samples, and dysfunctional monocytes and macrophages have been hypothesized to contribute to a protracted hyper-inflammatory state in COVID-19 patients. In this study, we demonstrate that CD169, a myeloid cell specific I-type lectin, facilitated ACE2-independent SARS-CoV-2 fusion and entry in macrophages. CD169-mediated SARS-CoV-2 entry in macrophages resulted in expression of viral genomic and subgenomic RNAs with minimal viral protein expression and no infectious viral particle release, suggesting a post-entry restriction of the SARS-CoV-2 replication cycle. Intriguingly this post-entry replication block was alleviated by exogenous ACE2 expression in macrophages. Restricted expression of viral genomic and subgenomic RNA in CD169[+] macrophages elicited a pro-inflammatory cytokine expression (TNFα, IL-6 and IL-1β) in a RIG-I, MDA-5 and MAVS-dependent manner, which was suppressed by remdesivir treatment. These findings suggest that *de novo* expression of SARS-CoV-2 RNA in macrophages contributes to the pro-inflammatory cytokine signature and that blocking CD169-mediated ACE2 independent infection and subsequent activation of macrophages by viral RNA might alleviate COVID-19-associated hyperinflammatory response.

## Author summary

Over-exuberant production of pro-inflammatory cytokine expression by macrophages has been hypothesized to contribute to severity of COVID-19 disease. Molecular

**Funding:** This work was supported by NIH grants R01AI064099 (SG), R01DA051889 (SG), R01AG060890 (SG), P30AI042853 (SG), R01CA227292 (ST), R01AI106036 (YB and ST), R01AI133486 (EM), and R21AI135912 (EM) as well as Fast Grants (EM) and Evergrande MassCPR (EM). The funders had no role in study design, data collection and analysis, decision to publish, or preparation of the manuscript.

**Competing interests:** The authors have declared that no competing interests exist.

mechanisms that contribute to macrophage-intrinsic immune activation during SARS-CoV-2 infection are not fully understood. Here we show that CD169, a macrophage-specific sialic-acid binding lectin, facilitates abortive SARS-CoV-2 infection of macrophages that results in innate immune sensing of viral replication intermediates and production of proinflammatory responses. We identify an ACE2-independent, CD169-mediated endosomal viral entry mechanism that results in cytoplasmic delivery of viral capsids and initiation of virus replication, but absence of infectious viral production. Restricted viral replication in CD169[+] macrophages and detection of viral genomic and subgenomic RNAs by cytoplasmic RIG-I-like receptor family members, RIG-I and MDA5, and initiation of downstream signaling via the adaptor protein MAVS, was required for innate immune activation. These studies uncover mechanisms important for initiation of innate immune sensing of SARS-CoV-2 infection in macrophages, persistent activation of which might contribute to severe COVID-19 pathophysiology.

## Introduction

Severe acute respiratory syndrome coronavirus 2 (SARS-CoV-2) is the causative agent of the COVID-19 pandemic. Severe COVID-19 cases have been associated with aberrant bronchioalveolar immune cell activation and persistently high levels of proinflammatory cytokines, including IL-6, TNFα, and IL-1β [1,2]. This protracted immune hyperactivation state marked by uncontrolled proinflammatory cytokine expression [3–6] is a potential driver of acute respiratory distress syndrome (ARDS) in severe COVID-19. Transcriptomic analysis of bronchioalveolar lavage fluid (BALF) samples from SARS-CoV-2 infected individuals revealed extensive lung infiltration by inflammatory monocytes and activated tissue-resident and BALF-associated macrophages with robust induction of interferon-stimulated gene (ISG) expression [7], suggestive of a myeloid cell-intrinsic cytokine signature contributing to ARDS and COVID-19 pathologies [8,9]. While numerous molecular mechanisms have been proposed to contribute to a myeloid cell-intrinsic hyperinflammatory phenotype [6,10–15], whether SARS-CoV-2 can establish productive infection in monocytes and macrophages has remained contentious [10,16–20].

  Studies on post-mortem tissues from patients, who succumbed to COVID-19, showed that a subset of tissue-resident alveolar macrophages are enriched in SARS-CoV-2 RNA [21,22]. Additionally, single-cell RNA-seq analysis revealed the presence of viral mRNAs in inflammatory myeloid cell populations in autopsied lung tissues [23,24]. Recent studies suggest that tissue-resident human macrophages are permissive to SARS-CoV-2 infection in humanized mice models [15], and that inhibition of viral genome replication or type-I interferon (IFN) signaling significantly attenuates chronic macrophage hyperactivation and disease progression [15]. However, whether the presence of viral RNA in macrophages reflects phagocytosis of infected bystander cells or active virus replication in tissue-resident macrophages has yet to be clearly defined. In contrast, CD14[+] peripheral blood monocytes, monocyte-derived dendritic cells (MDDCs), or monocyte-derived macrophages (MDMs) were not-permissive to productive SARS-CoV-2 replication in vitro [16–18,20]. In permissive lung epithelial cells and those expressing the cognate entry receptor, angiotensin-converting enzyme 2 (ACE2), SARS-CoV-2 utilizes its spike (S) glycoprotein to interact with ACE2, which facilitates proteolytic cleavage, plasma or endosomal membrane fusion, and cytosolic import of viral genome [25–27]. Depending on cell type, different host proteases such as furin, TMPRSS2, or cathepsins are required for S cleavage and entry of SARS-CoV-2 [26,28,29]. While circulating monocytes and

macrophages are not known to express ACE2 [30], these cells have been shown to express low levels of endogenous surface TMPRSS2 [31], and moderate levels of endosomal cathepsins [32,33], although the relative expression of these cellular proteases in the context of SARS-CoV-2 infection and inflammation is not well understood.

Recent reports have highlighted capture of SARS-CoV-2 virus particles by myeloid cell-specific receptors, such as C-type lectins, or entry of antibody-opsonized virus particles by FcγRs in an ACE2-independent manner, though productive viral infection was not observed [10–12,14,15,34,35]. We and others have previously shown that CD169/Siglec-1 facilitates viral infections of macrophages or dendritic cell (DC)-mediated trans infection of bystander cells [36–39]. CD169 binds to sialylated viral glycoproteins or viral membrane-associated gangliosides, GM1 and GM3 [38–44]. SARS-CoV-2 S is extensively glycosylated with some of the complex glycans having terminal sialic acid residues [45,46], and a recent report demonstrated that DC-mediated SARS-CoV-2 trans infection of ACE2$^+$ epithelial cells was facilitated by CD169 [14]. CD169 is highly expressed by splenic red pulp and perifollicular macrophages, subcapsular sinus macrophages [47] and alveolar macrophages [48,49]. Besides constitutive expression on tissue-resident macrophages, CD169 expression can be upregulated on peripheral blood monocytes under inflammatory conditions, especially in response to type I interferons (IFNs) [50–52]. Since CD169 expression is elevated on peripheral blood monocytes, interstitial and alveolar macrophages in COVID-19 patients [21,22,53,54], we reasoned that SARS-CoV-2 S mediated interactions with CD169$^+$ macrophages might play a crucial role in driving immunopathology of SARS-CoV-2 infection.

In this study, we examined the role of CD169 in facilitating SARS-CoV-2 infection of ACE2-deficient human macrophages and its effect on inducing pro-inflammatory cytokine expression. Using two different human macrophage models, PMA-differentiated THP1 cells (THP1/PMA) and primary monocyte-derived macrophages (MDMs), we show that CD169 binds to SARS-CoV-2 S and mediates SARS-CoV-2 S-dependent viral entry into macrophages, leading to restricted cytosolic expression of viral genomic and subgenomic (sg) RNA. Surprisingly, induced constitutive expression of ACE2 in macrophages (THP1/PMA and MDMs) restored permissiveness to robust virus replication and production of infectious progeny virions, suggesting that ACE2 expression overcomes a post-entry block to SARS-CoV-2 infection in macrophages. While CD169-mediated, ACE2-independent SARS-CoV-2 entry into macrophages resulted in negligible viral protein expression and absence of infectious virus production, restricted expression of viral negative strand RNA and sgRNAs induced pro-inflammatory cytokine expression via retinoic acid-inducible gene I (RIG-I) and melanoma differentiation-associated gene 5 (MDA-5) dependent sensing of viral replication intermediates. Importantly, expression of IL-6, TNFα and IL-1β was enhanced, whereas type I IFN responses were muted, suggesting a novel CD169-mediated, macrophage-intrinsic amplification of pro-inflammatory responses. These findings suggest that induction of pro-inflammatory responses in SARS-CoV-2-exposed macrophages requires initial viral RNA synthesis and that abortively infected macrophages might contribute to the hyperimmune phenotype and pathophysiology of COVID-19.

## Results

### SARS-CoV-2 Spike protein can mediate ACE2-independent entry into macrophages

To examine the role of macrophages in SARS-CoV-2 infection and COVID-19 pathogenesis, we differentiated primary MDMs from multiple donors by culturing CD14$^+$ monocytes in the presence of human AB-serum and M-CSF for 6 days [55]. Compared to a control HEK293T/

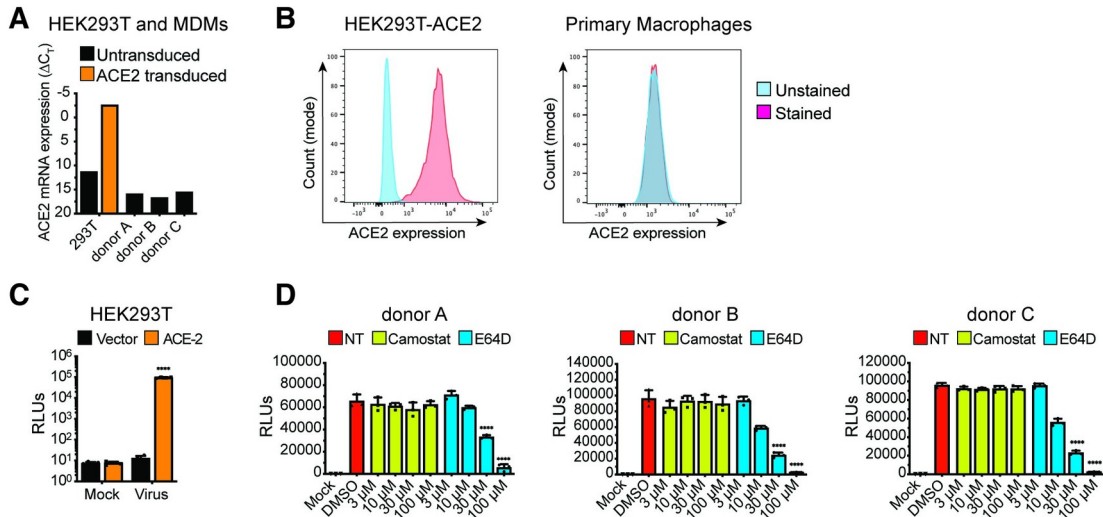

**Fig 1. ACE2-independent SARS-CoV-2 entry in macrophages.** (**A**-**B**) Representative ACE2 mRNA expression (**A**) and flow cytometry profiles showing ACE2 surface expression (**B**) of HEK293T cells stably expressing ACE2 and primary MDMs from multiple donors. (**C**-**D**) Parental (vector) and transduced (ACE2) HEK293T cells (**C**) and primary MDMs from 3 donors (**D**) were infected with S-pseudotyped lentivirus (20 ng based on p24$^{Gag}$), in the absence or presence of cathepsin inhibitor (E64D) or TMPRSS2 inhibitor (Camostat), and infection was quantified by measuring luciferase activity at day 3 post-infection. Mock: no virus added, DMSO: no-treatment. The means ± SEM from at least 3 independent experiments are shown. *P*-values: paired t-test, two-tailed comparing to vector control (**C**), or one-way ANOVA followed by the Dunnett's post-test comparing to DMSO (**D**). ****: $p < 0.0001$.

ACE2 cell line retrovirally transduced to stably express human ACE2, ACE2 mRNA expression in primary human MDMs was very low, and we were unable to detect surface expression of ACE2 protein on MDMs (**Fig 1A** and **1B**). However, similar to HEK293T/ACE2 cells (**Fig 1C**), MDMs from multiple donors were robustly infected with a SARS-CoV-2 S-pseudotyped lentivirus (**Fig 1D**), suggesting that macrophages can support ACE2-independent S-pseudotyped virus entry. Furthermore, SARS-CoV-2 S-pseudotyped infections in MDMs were blocked by treatment with a cathepsin inhibitor (E64D) but not a TMPRSS2 inhibitor (Camostat) (**Fig 1D**), suggesting that SARS-CoV-2 S facilitates endosomal viral entry into ACE2-deficient MDMs. Preferential engagement of endosomal entry mechanism for SARS-CoV-2 S pseudotyped lentiviruses (LVs) in MDMs correlated with lack of TMPRSS2 expression in THP1/PMA and primary macrophages ([56], **S1A Fig**), and robust cathepsin-L expression in THP1/PMA and MDMs (**S1B Fig**). Strikingly, pre-treatment with neutralizing antibodies targeting the N-terminal domain (NTD) of SARS-CoV-2 S that do not compete with ACE2 binding by S [57], led to significant reduction in S-pseudotyped lentivirus infection of primary MDMs, suggesting that specific interaction between SARS-CoV-2 S and ACE2-independent entry factors are essential to mediate entry and endosomal fusion in macrophages (**S2 Fig**).

## CD169 is a SARS-CoV-2 attachment and entry factor in macrophages

Expression of CD169, an ISG, is significantly upregulated in monocytes and alveolar macrophages isolated from COVID-19 patients, and its expression enhancement correlates with COVID-19 disease severity [7,53]. While co-expression of CD169 with ACE2 can enhance ACE2-mediated SARS-CoV-2 S entry in HEK293T cells [11], whether CD169 plays a role in SARS-CoV-2 infection of ACE2-deficient macrophages is unclear. To address this question, THP1 cells stably expressing wildtype (wt) CD169, mutant CD169/R116A which is defective in its ability to bind sialylated glycoconjugates [39,58,59], ACE2, or both wt CD169 and ACE2

(CD169/ACE2) (**S3A Fig**) were incubated with recombinant SARS-CoV-2 S protein, and relative S binding was determined by flow cytometry. In contrast to parental THP1 monocytes, THP1 cells expressing wt CD169 (THP1/CD169) displayed robust S binding comparable to levels observed with THP1/ACE2 cells (**Fig 2A**). There was a significant reduction in S binding to THP1/CD169-R116A cells, indicating that the sialylated S protein is recognized by CD169 (**Fig 2A**). Co-expression of CD169 and ACE2 in THP1 cells further enhanced S binding compared to cells expressing only ACE2 or CD169, suggestive of cooperative binding of CD169 and ACE2 to SARS-CoV-2 S.

We next sought to determine the role of CD169 in mediating SARS-CoV-2 infection of macrophages. PMA-differentiated THP1 macrophages expressing wt or mutant CD169, ACE2, or CD169/ACE2 were infected with SARS-CoV-2 S-pseudotyped lentiviruses. Expression of wt CD169 on THP1/PMA macrophages enhanced Wuhan S-pseudotyped lentiviral infection by ~30-fold compared to parental THP1/PMA macrophages, similar to the levels of infection observed with ACE2+ THP1/PMA macrophages (**Fig 2B**). In contrast, expression of CD169/R116A did not enhance S-pseudotyped lentiviral infection of THP1/PMA cells, confirming that recognition of sialylated motifs on SARS-CoV-2 S is essential for CD169-mediated infection of macrophages (**Fig 2B**). Co-expression of wt CD169 and ACE2 further enhanced S-pseudotyped lentiviral infection by greater than 11-fold and 3-fold when compared to cells expressing CD169 or ACE2, respectively (**Fig 2B**), confirming that CD169 facilitates S-mediated entry into macrophages in the absence of ACE2 and enhances entry in the presence of ACE2. Notably, CD169-mediated entry enhancement of SARS-CoV-2 was also observed with delta (**Fig 2C**) and omicron (**Fig 2D**) S-pseudotyped lentiviruses in THP1/PMA macrophages.

To confirm the role of CD169 in SARS-CoV-2 S-mediated infection in primary human MDMs, and given the relatively low and variable expression of endogenous CD169 in primary MDMs (**S3B Fig**), we used lentiviral transduction to overexpress either wt CD169 or ACE2 (**S3C** and **S3D Fig**). CD169 or ACE2 overexpression in MDMs from multiple donors led to a significant increase in SARS-CoV-2 (Wuhan) S-pseudotyped lentivirus infection compared to untransduced MDMs or those transduced with empty vector (**Fig 2E**). Crucially, pre-treatment with anti-CD169 blocking mAb (7D2) prior to infection with S-pseudotyped lentivirus significantly attenuated infection of untransduced primary MDMs when compared to pre-treatment with IgG (**Fig 2F**). These findings suggest that CD169 facilitates SARS-CoV-2 S-dependent fusion and entry into both THP1/PMA macrophages and primary MDMs in the absence of ACE2.

## SARS-CoV-2 establishes abortive infection in macrophages lacking ACE2

To investigate whether CD169 expression is sufficient to establish productive SARS-CoV-2 infection and replication in ACE2-deficient macrophages, we infected THP1/PMA overexpressing CD169, ACE2, or parental cells (lacking both CD169 and ACE2) with replication-competent SARS-CoV-2 (Washington isolate, NR-52281), as previously described [60]. To evaluate productive infection, we examined the temporal presence of double-stranded RNA (dsRNA), a viral replication intermediate [61], as well as viral nucleocapsid (N) protein expression. SARS-CoV-2-infected THP1/PMA macrophages were fixed at various time points post infection and subjected to immunofluorescence analysis using antibodies against dsRNA and SARS-CoV-2 N. In contrast to parental THP1/PMA macrophages that showed background staining of dsRNA and no expression of SARS-CoV-2 N at any time point post infection (**Fig 3A**), CD169-expressing THP1/PMA cells displayed low levels of dsRNA production and small puncta staining of SARS-CoV-2 N that did not significantly increase over the course of

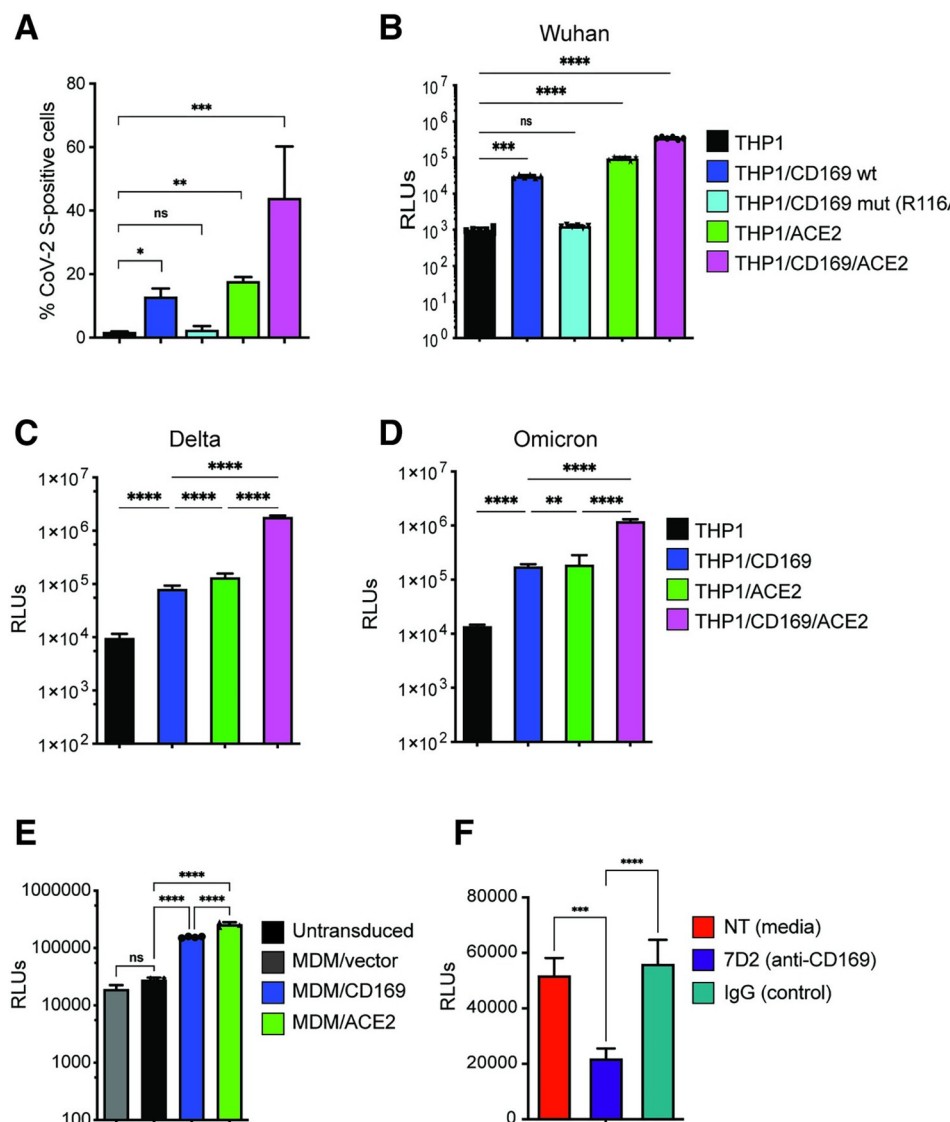

**Fig 2. CD169 is a SARS-CoV-2 attachment and entry factor in macrophages.** (**A**) Binding of SARS-CoV-2 S protein (Wuhan isolate) to THP1 monocytes expressing wt CD169, mutant CD169 (R116A), ACE2, or both wt CD169 and ACE2 (CD169/ACE2). (**B**) THP1/PMA macrophages were infected with Wuhan (**B**), Delta (**C**) or Omicron (**D**) S-pseudotyped lentivirus (20 ng p24$^{Gag}$). Infection was quantified by measuring luciferase activity at 3 dpi. Relative light units (RLUs) from each cell line were normalized to no virus control (mock). The means ± SEM are shown from at least five independent experiments. (**E**) Primary MDMs from three donors overexpressing either CD169 or ACE2, or vector control were infected with Wuhan S-pseudotyped lentivirus (20 ng p24$^{Gag}$) for 3 days, followed by analysis of luciferase activity in whole cell lysates. (**F**) Untransduced primary MDMs (representative of 3 donors) were pre-treated with anti-CD169 mAb (20 μg/ml, 7D2), IgG1, or left untreated (NT) for 30 min at 4°C prior to infection with Wuhan S-pseudotyped lentivirus (20 ng p24$^{Gag}$) for 3 days, followed by analysis of luciferase activity. RLUs from each donor in each group were normalized to no virus control (mock). The means ± SEM from 3 independent experiments are shown. *P*-values: paired t-test, two-tailed (**A**), one-way ANOVA followed by the Dunnett's post-test (**B**) or Tukey's post-test comparing to untransduced cells (**C-E**), or each pre-treatment condition (**F**). **: $p < 0.01$; ***: $p < 0.001$; ****: $p < 0.0001$; ns: not significant.

infection (**Fig 3B**). However, both ACE2$^+$ (**Fig 3C**) and CD169/ACE2 double-positive (**Fig 3D**) THP1/PMA macrophages showed robust dsRNA and SARS-CoV-2 N antigen production starting as early as 2–4 hours post infection (hpi), with substantial increases in N expression over 24 hpi (**Fig 3C** and **3D**). There was a clear distinction in the spatial distribution of viral N

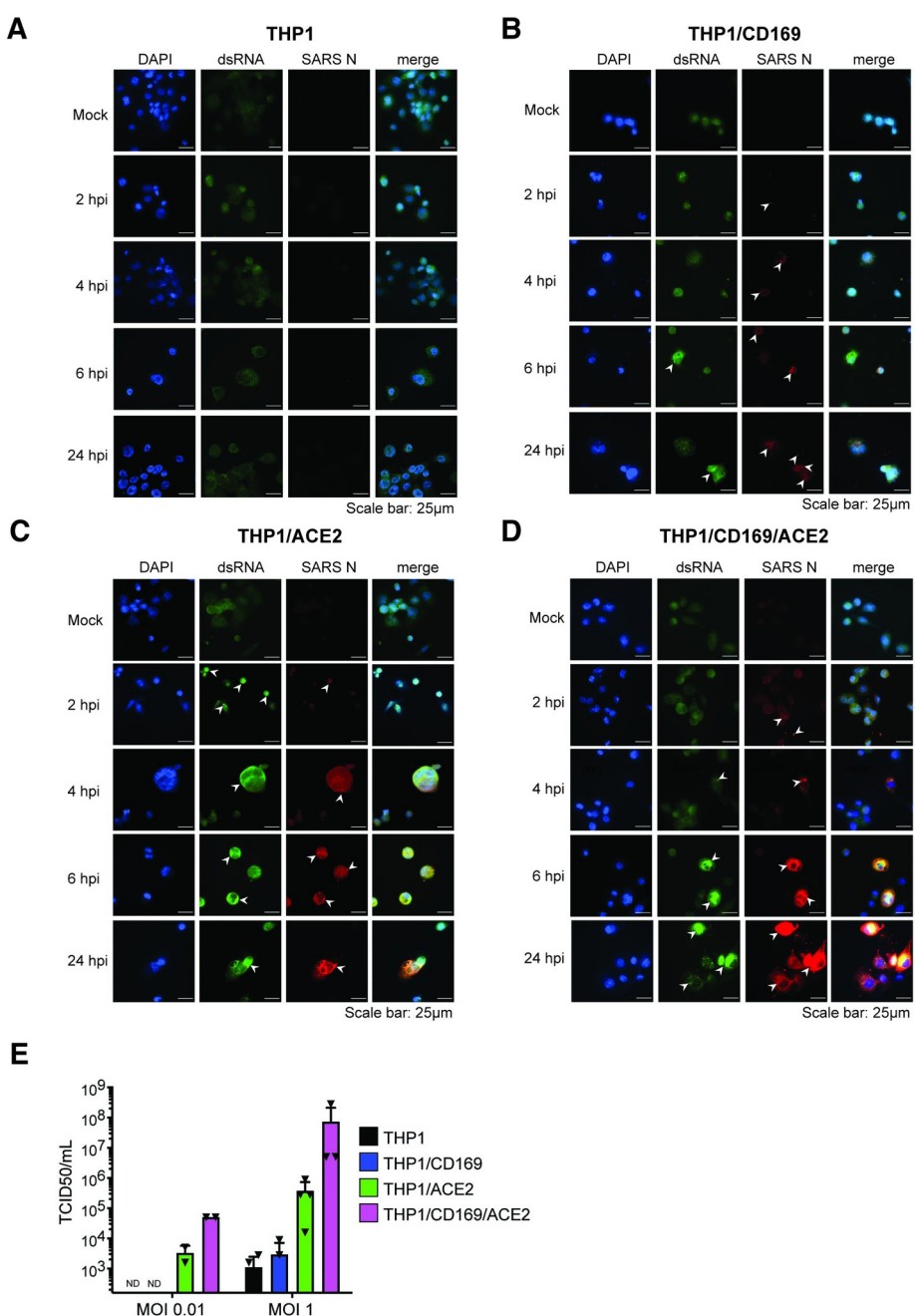

**Fig 3. SARS-CoV-2 establishes abortive infection in CD169$^+$ THP1/PMA macrophages.** (**A-D**) Representative immunofluorescence images (100x) of THP1/PMA macrophages infected with SARS-CoV-2 (MOI = 1) and stained for nucleus (DAPI, blue), SARS-CoV-2 dsRNA (green), and SARS-CoV-2 nucleocapsid (N, red), at indicated times post infection. Images shown for each cell line; untransduced control (parental, **A**), CD169+ (**B**), ACE2+ (**C**), and CD169+/ACE2+ (**D**). Bar = 25 μm. (**E**) Culture supernatants from SARS-CoV-2 infected THP1/PMA macrophages were harvested at 24 hpi and viral titers determined by TCID$_{50}$ assay. The means ± SEM from 3 independent experiments are shown.

protein in infected ACE2$^+$ and CD169$^+$/ACE2$^+$ THP1/PMA macrophages over time, as noted by a transition from 6 hpi onward from small cytosolic N protein puncta to homogenous distribution throughout the cytosol (**Fig 3C** and **3D**). Furthermore, there was extensive co-localization between peri-nuclear dsRNA foci and N staining in ACE2$^+$ and CD169$^+$/ACE2$^+$ THP1/PMA macrophages at 6 and 24 hpi. In accordance with our S-pseudotyped lentivirus infection data (**Fig 2B**), co-expression of CD169 and ACE2 led to an increase in SARS-CoV-2 infection compared to ACE2-expressing cells, with higher levels and slightly earlier production of dsRNA and SARS-CoV-2 N (compare **Fig 3C** and **3D**). In contrast, N protein staining was not dispersed in the cytoplasm of CD169$^+$ THP1/PMA macrophages and remained in small cytosolic puncta over the course of infection with minimal co-localization with dsRNA staining (**Fig 3B**). Similar to the findings in THP1/PMA macrophages, exogenous expression of CD169 in HEK293T cells resulted in significant enhancement of SARS-CoV-2 S pseudotyped lentivirus entry and infection (**S4A Fig**), but viral protein expression was not observed in HEK293T/CD169 cells infected with recombinant SARS-CoV-2 expressing NeonGreen (SARS-CoV-2-mNG) (**S4B Fig**). We further quantified infectious SARS-CoV-2 particle production from THP1/PMA macrophages (**Fig 3E**) by TCID$_{50}$ assay and observed robust viral particle release only in ACE2-expressing cells, with a marked increase in viral titers in cells co-expressing both CD169 and ACE2 compared to ACE2 alone (**Fig 3E**). These findings suggest that in absence of ACE2, CD169-mediated SARS-CoV-2 entry in THP1/PMA macrophages is restricted to initiation of viral transcription and low levels of viral protein production.

## CD169-mediated SARS-CoV-2 infection of macrophages results in *de novo* expression of viral transcripts

To explore the spatial and temporal distribution of SARS-CoV-2 RNA at single cell level, single-molecule RNA FISH (smFISH) analysis [62] was performed in infected THP1/PMA macrophages. Individual fluorescent spots corresponding to viral genomic RNA (gRNA) and N mRNAs were detected in CD169$^+$, ACE2$^+$ and CD169$^+$/ACE2$^+$ THP1/PMA macrophages, as early as 1 hpi (**Fig 4A**). By 6 hpi, we observed diffuse cytosolic staining and perinuclearly localized bright gRNA foci (**Fig 4A**), which further increased in ACE2$^+$ and CD169$^+$/ACE2$^+$ but not CD169$^+$ THP1/PMA macrophages (**Fig 4A**). In order to distinguish between gRNAs and N gene transcripts, we infected cells for 24 hours and stained them with two probe sets labeled with distinguishable dyes one against SARS-CoV-2 ORF1a and the other against N. The former detects only gRNA and the later binds both gRNA and N subgenomic transcripts (**Fig 4B**). The staining, particularly that of N gene, showed diffused cytoplasmic distribution, only in ACE2$^+$ and CD169$^+$/ACE2$^+$ THP1/PMA macrophages, especially in the advanced stages of infection (**Fig 4B**). Dispersion of N transcripts (gRNA and sgRNA) in the cytosol of ACE2$^+$ and CD169$^+$/ACE2$^+$ THP1/PMA macrophages mirrored the temporal localization phenotype of viral N protein (**Fig 3C** and **3D**), and might be indicative of formation of N positive viral replication compartments [63].

While a majority of virus-exposed CD169$^+$ THP1/MA macrophages expressed fluorescent puncta (gRNA) at 24 hpi (indicated by the white arrowheads showing both ORF1a and N expression, **Fig 4B**), transition to bright gRNA foci or cytosolic expansion was not observed, suggesting that CD169 expression in THP1/PMA cells increased uptake of virus, but without ACE2, failed to establish viral replication foci. It should be noted that the formation of distinct fluorescent puncta in CD169$^+$ THP1/PMA macrophages at 24 hpi is suggestive of *de novo* RNA synthesis and not virus inoculum, since these puncta were not detected at 1 hpi (**Fig 4A**). While viral RNAs in the CD169-expressing cells were localized in few distinct granulated puncta, SARS-CoV-2 infection of ACE2-expressing cells led to the formation of viral RNA-

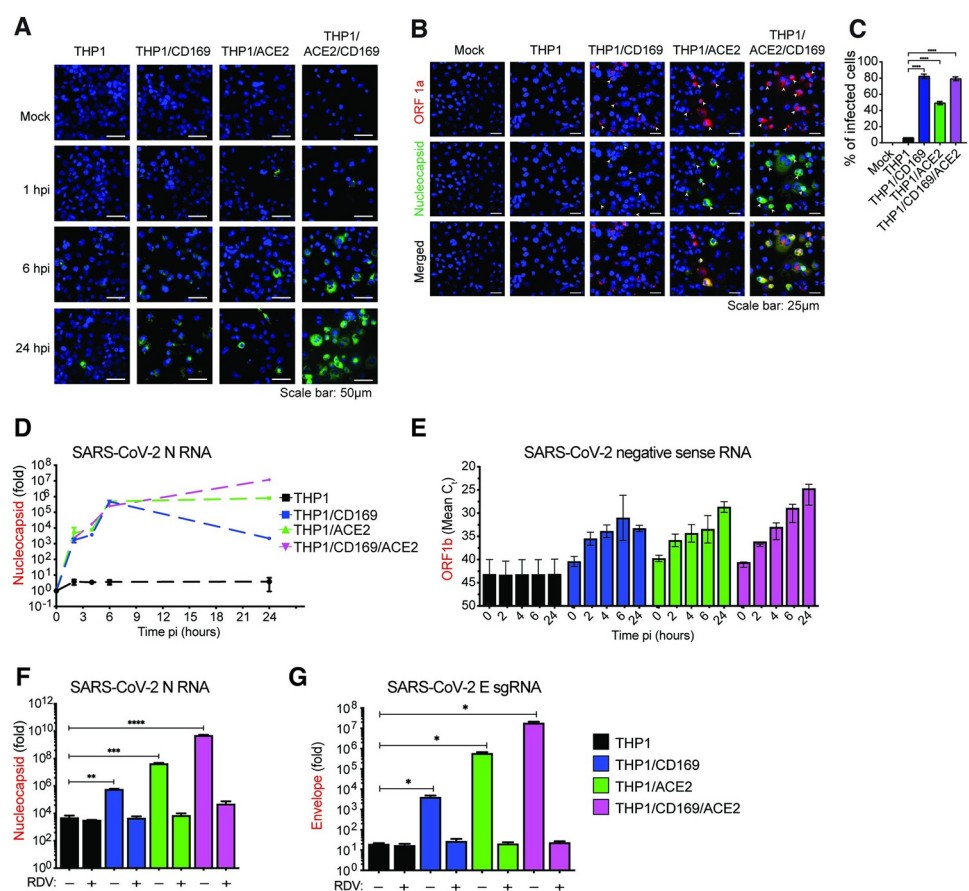

**Fig 4. CD169-mediated SARS-CoV-2 infection results in restricted viral RNA expression in THP1/PMA macrophages.** (**A**) Single molecule RNA FISH analysis of viral positive sense gRNA using high fidelity probes in SARS-CoV-2 infected THP1/PMA macrophages (MOI = 10) at the indicated timepoints. Representative fields of cells were hybridized at indicated times with 7 sets of smFISH probes labeled with Quasar670 targeting the + strand of SARS-CoV-2 ORF1a (NSP1-3) and N transcripts. Data are representative of 2 independent experiments. Bar = 50 μm. (**B**) smFISH analysis of viral RNAs using ORF1a specific probe set (labeled with Texas Red) and N specific probe set (labelled with TMR) in SARS-CoV-2 infected THP1/PMA cells (MOI = 10, 24 hpi). Data are representative of 2 independent experiments. Bar = 25 μm. (**C**) Percentage of infected cells based on the presence of SARS-CoV-2 RNAs determined from 10–20 independent fields (representative field shown in **B**). (**D**-**G**) THP1/PMA macrophages infected with SARS-CoV-2 (MOI = 1) in the absence (**D**-**E**) or presence (**F**-**G**) of remdesivir (RDV, 1 μM). Total RNA was harvested at indicated times post-infection and expression of viral transcripts was quantified by RT-qPCR. Replication kinetics of SARS-CoV-2 were quantified by (**D**) total N RNA amplification (values from each group normalized to mock (uninfected) or (**E**) negative sense antigenome from strand-specific reverse-transcription and ORF1b amplification (mean $C_t$ values for each condition). Expression of total N RNA (**F**) and E sgRNA (**G**) transcripts in THP1/PMA macrophages was analyzed at 24 hpi. Values from each group normalized to mock (uninfected). The means ± SEM are shown from at least 3 independent experiments. Significant differences between groups were determined by one-way ANOVA followed by the Dunnett's post-test (**C, F**-**G**), comparing to parental THP1 cells. *P*-values: $^{*}<0.1$; $^{**}<0.01$; $^{***}<0.001$; $^{****}<0.0001$.

containing inclusion-like structures, suggesting that differential engagement of viral entry receptors (such as CD169 and ACE2) leads to altered fate of the incoming viral genome and subsequent steps in the viral replication cycle.

To further investigate the step at which SARS-CoV-2 replication is restricted, THP1/PMA macrophages or those expressing either CD169, ACE2, or both CD169 and ACE2 were infected with SARS-CoV-2 and lysed at 2, 4, 6, and 24 hpi to quantify the level of total

SARS-CoV-2 N transcripts, positive sense gRNA and sgRNAs, as well as negative sense antigenomic RNA, by RT-qPCR. We detected increasing levels of total SARS-CoV-2 N transcripts at early time points (2–6 hpi) in THP1/PMA macrophages expressing CD169, ACE2, or both CD169 and ACE2, whereas parental THP1/PMA macrophages showed no significant increase in viral N RNA levels compared to mock infection controls (**Fig 4D**). Furthermore, there were no significant differences in total N RNA levels at early times post infection (up to 6 hpi) between CD169[+], ACE2[+] or CD169[+]/ACE2[+] THP1/PMA macrophages suggesting an absence of cell-intrinsic restrictions to early steps of SARS-CoV-2 replication, such as attachment and fusion, in CD169+ THP1/PMA macrophages. Interestingly, SARS-CoV-2 N transcripts peaked at 6 hpi in CD169+ THP1/PMA cells, followed by a progressive decline at 24 hpi (**Fig 4D**). In contrast, there was an increase in SARS-CoV-2 N transcripts over time in cells expressing ACE2, with markedly higher expression at 24 hpi compared to CD169-expressing cells. In concordance with SARS-CoV-2 N protein expression (**Fig 3D**), THP1/PMA cells expressing both CD169 and ACE2 expressed the highest levels of SARS-CoV-2 N transcripts at 24 hpi compared to those expressing ACE2 or CD169 alone.

After virus entry and uncoating, the next steps in the SARS-CoV-2 replication cycle are translation of the viral gRNA followed by the formation of viral replication-transcription complexes, which enables synthesis of the negative sense antigenomic RNA [64]. Negative sense antigenomic RNAs are present at significantly lower levels than positive sense gRNA and sgRNAs, and are templates for synthesis of additional positive sense gRNA and sgRNAs [64]. We employed strand-specific RT-qPCR analysis to detect negative sense viral RNA (ORF1b) and confirmed viral replication (at 2 hpi) in CD169[+] THP1/PMA cells, compared to complete absence of negative sense viral RNAs in parental THP1/PMA macrophages (**Fig 4E**). Interestingly, negative sense antigenomic RNA expression in CD169[+] THP1/PMA macrophages plateaued at 6 hpi, suggesting that CD169-mediated SARS-CoV-2 entry only promotes initial steps of viral replication. In contrast, there was a progressive increase in negative sense antigenomic RNA levels in ACE2[+] THP1/PMA macrophages particularly at 24 hpi, which was even more pronounced in cells co-expressing CD169 and ACE2 (**Fig 4E**). To confirm that the increase in viral transcripts in CD169[+] or ACE2[+] THP1/PMA macrophages was due to ongoing virus transcription, THP1/PMA cells (±CD169 ±ACE2) were treated with remdesivir (RDV) [15,63]. RT-qPCR analysis of both total N RNA (**Fig 4F**) and sgRNA (E transcripts) (**Fig 4G**) harvested at 24 hpi revealed that RDV treatment completely blocked the increase in viral RNA expression in THP1/PMA macrophages expressing CD169, ACE2, or both CD169 and ACE2, to levels observed in parental THP1/PMA macrophages.

To further validate these findings in primary macrophages, we transduced MDMs from multiple donors with empty vector (negative control) or lentiviruses overexpressing either CD169 or ACE2, and infected the cells with SARS-CoV-2. Note that MDMs constitutively express CD169 and expression of CD169 was only moderately induced upon lentivector transduction (**S3B Fig**). Similar to findings with CD169[+] THP1/PMA macrophages, smFISH analysis to detect Orf1a transcripts revealed diffuse cytoplasmic staining with isolated perinuclear gRNA foci in untransduced or CD169-overexpressing infected MDMs at 24 hpi (**Fig 5**). Notably, there was no significant difference in numbers of Orf1a gRNA+ cells in CD169-overexpressing MDMs compared to untransduced MDMs (**Fig 5B**). In contrast, ACE2 overexpression led to a significant enhancement in perinuclear Orf1a staining (**Fig 5A** and **5B**), suggestive of robust virus replication in ACE2+ MDMs. While both N RNAs and E sgRNAs were robustly expressed in SARS-CoV-2 infected ACE2[+] MDMs, RNA expression levels were only slightly higher in CD169[+] MDMs compared to untransduced MDMs (**Fig 5C** and **5D**). Similarly, N protein expression and infectious virus particle production were only observed in ACE2[+] MDMs but not in CD169[+] or untransduced MDMs at 24 hpi (**Fig 5E** and

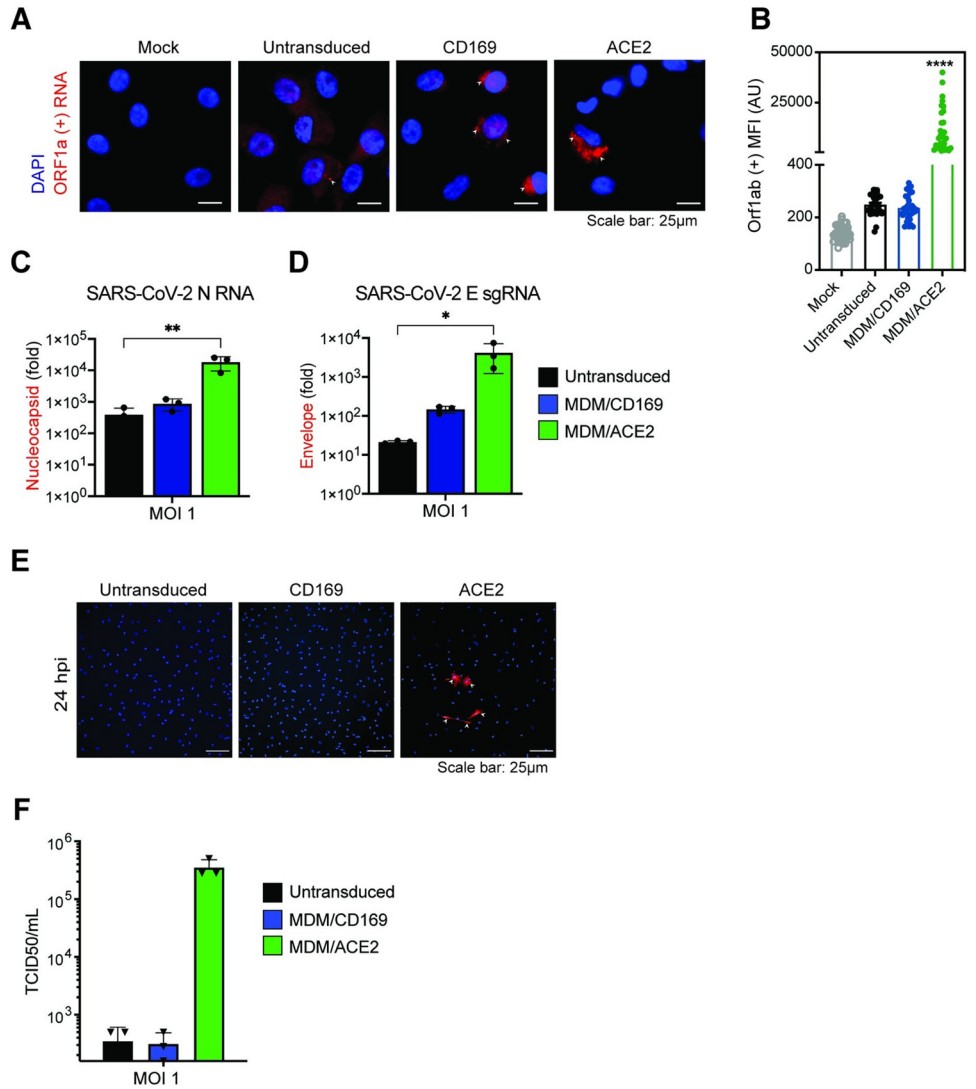

**Fig 5. CD169-mediated virus entry leads to restricted SARS-CoV-2 infection in primary MDMs.** (**A**) Expression of SARS-CoV-2 genomic RNA in primary MDMs imaged using smFISH probes against Orf1a (NSP 1–3), and DAPI for nuclear staining. Data are a representative of 2 independent experiments. Bar = 25 μm. (**B**) Quantitation of mean fluorescence intensities (MFI) based on the presence of SARS-CoV-2 gRNAs determined from 10–20 independent fields. Significant differences between the groups were determined by one-way ANOVA showing mean fluorescence intensity ± SEM for each group. (**C-D**) Total RNA from SARS-CoV-2 infected primary untransduced MDMs (MOI = 1, 24 hpi), or MDMs overexpressing either CD169 or ACE2 was analyzed by RT-qPCR for expression of total N RNA (**C**), and envelope sgRNA (**D**). Fold expression normalized to mock (uninfected) control for each donor. Significant differences between groups were determined by one-way ANOVA followed by the Dunnett's post-test (**C-D**), comparing to untransduced control for each donor. *P*-values: *<0.1; **<0.01; ****<0.0001. (**E**) Representative immunofluorescence images (20x) from primary untransduced MDMs, or MDMs overexpressing either CD169 or ACE2, infected with SARS-CoV-2 (MOI = 1, 24 hpi) and stained for nucleus (DAPI, blue), and SARS-CoV-2 N (red). Bar = 25 μm. (**F**) Culture supernatants from SARS-CoV-2 infected primary MDMs from multiple donors were harvested at 24 hpi and viral titers determined by $TCID_{50}$ assay.

**5F**). Taken together, these results suggest that CD169-mediated viral entry enables initiation of SARS-CoV-2 RNA replication and transcription in ACE2-deficient CD169⁺ macrophages but does not allow for the establishment of productive infection, suggesting a restriction at a post-entry step of the viral replication cycle in macrophages.

## Low level expression of SARS-CoV-2 genomic and subgenomic RNAs is sufficient to induce pro-inflammatory cytokine expression in non-productively infected macrophages

Since CD169 expression in macrophages was sufficient to permit SARS-CoV-2 entry and initiate limited viral RNA production, we investigated whether *de novo* viral RNA synthesis in the absence of productive virus replication can trigger innate immune responses. SARS-CoV-2 infected THP1/PMA cells (±CD169 ±ACE2) were lysed at 24 hpi, and total RNA was analyzed by RT-qPCR for inflammatory cytokine expression (**Fig 6**). We observed significant induction

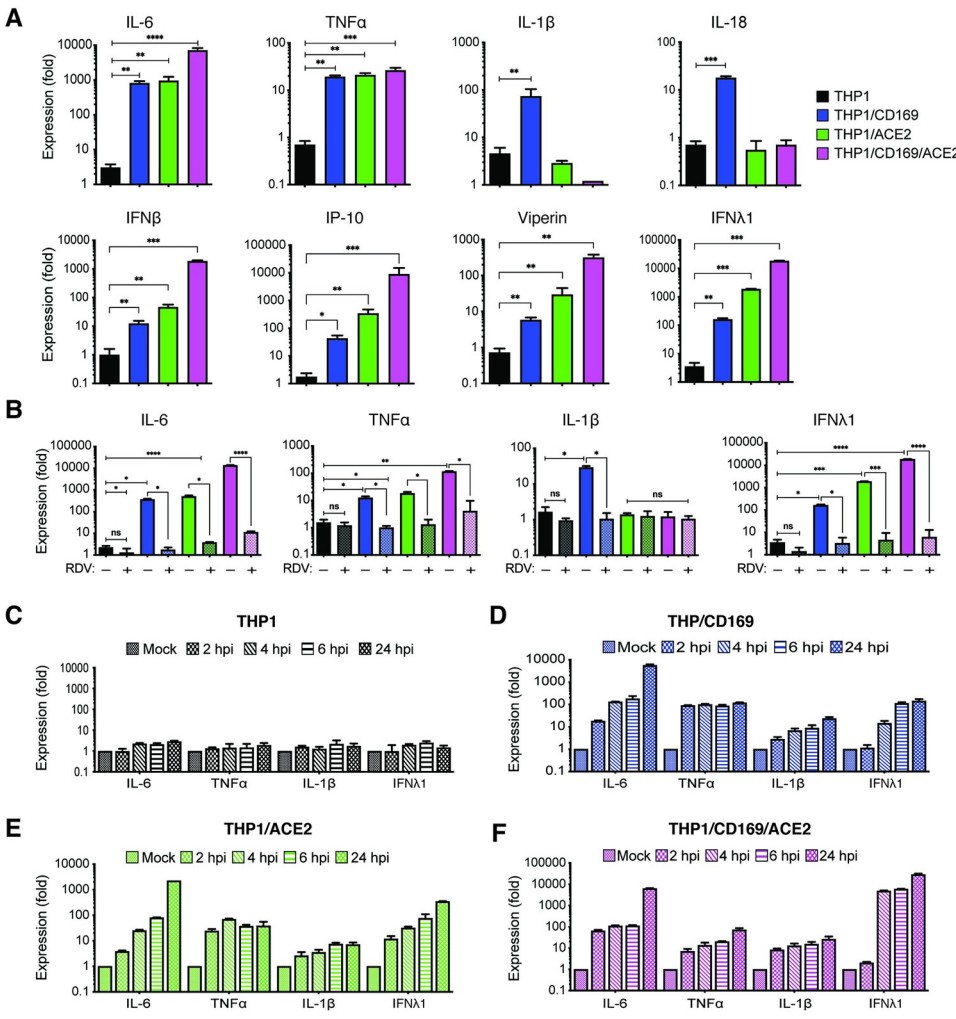

**Fig 6. Restricted de novo expression of SARS-CoV-2 RNA induces pro-inflammatory responses in non-productively infected THP-1/PMA macrophages.** (**A**-**B**) PMA-differentiated THP1 cells from parental (THP1) or those expressing CD169, ACE2, or CD169/ACE2 were infected with SARS-CoV-2 (MOI = 1, 24 hpi) in the absence or presence of RDV (1 μM), and total RNA was analyzed by RT-qPCR for pro-inflammatory cytokine and ISG expression. Values were normalized to mock-infected control for each group. Fold-induction for indicated pro-inflammatory cytokines (**A**, top panel) and ISGs (**A**, bottom panel) in the absence of RDV, or in the presence of RDV (**B**). (**C**-**F**) Kinetics of pro-inflammatory cytokine/ISG mRNA expression in the absence of RDV in parental (**C**), CD169-expressing (**D**), ACE2-expressing (**E**), and CD169/ACE2-expressing (**F**) THP1/PMA cells. The means ± SEM are shown from at least 3 independent experiments. Significant differences between groups were determined by one-way ANOVA followed by the Dunnett's post-test (**A**-**B**), comparing to control parental THP1 cells. *P*-values: *<0.1; **<0.01; ***<0.001; ****<0.0001; ns: not significant.

of IL-6, TNFα, IL-1β, IL-18, and ISGs in non-productively infected CD169$^+$ THP1/PMA macrophages compared to the parental THP1/PMA cells (**Fig 6A**). IL-6 and TNFα mRNA expression in SARS-CoV-2 infected THP1/CD169 cells was similar to the levels observed in productively infected THP1/ACE2 cells, but lower than that observed in THP1/CD169/ACE2 cells (**Fig 6A**, top panel). Intriguingly, induction of pro-inflammatory cytokines, IL-1β and IL-18, was only consistently observed in non-productively infected THP1/CD169 macrophages but not in the productively infected THP1/ACE2 or THP1/CD169/ACE2 macrophages. A delayed or impaired type I IFN response is thought to be a critical mechanism for COVID-19 pathogenesis, though impaired induction of type I IFN response has mostly been reported in infected lung epithelial cells [65]. Interestingly, we observed muted induction of IFNβ, IP-10 and Viperin in SARS-CoV-2 infected CD169$^+$ THP1/PMA macrophages (**Fig 6A**, bottom panel). In contrast, IFNβ, IFNλ1, IP-10 and Viperin mRNA expression was dramatically upregulated upon establishment of productive SARS-CoV-2 infection in THP1/ACE2 and THP1/CD169/ACE2 macrophages (**Fig 6A**) and correlated with the extent of viral RNA expression and virus replication (**Figs 3** and **4**).

Since previous reports have suggested that macrophage exposure to recombinant S protein may trigger inflammatory cytokine production [12,66], we infected parental, CD169$^+$, ACE2$^+$, or CD169$^+$/ACE2$^+$ THP1/PMA cells with Wuhan S-pseudotyped lentivirus. However, we observed no induction of pro-inflammatory cytokines (IL-6, TNFα, IL-1β, and IFNλ1) in S-pseudotyped lentivirus-infected cells (**S5 Fig**). To confirm that the induction of pro-inflammatory cytokine and ISG expression was due to virus replication and sensing of viral RNA, we infected THP1/PMA macrophages with SARS-CoV-2 in the presence of RDV. Treatment with RDV which inhibits viral RNA synthesis (**Fig 4F** and **4G**) completely abrogated IL-6, TNFα, IL-1β and IFNλ1 induction in CD169$^+$, ACE2$^+$, and CD169$^+$/ACE2$^+$ THP1/PMA macrophages (**Fig 6B**), suggesting that *de novo* viral RNA synthesis and sensing of viral replication intermediates are required for the induction of pro-inflammatory cytokines and ISGs in macrophages. Temporal analysis of inflammatory cytokine induction revealed that while SARS-CoV-2 infection of parental THP1/PMA cells did not result in induction of pro-inflammatory responses (**Fig 6C**), infection of CD169$^+$ THP1/PMA cells led to rapid induction of IL-6, TNFα, IL-1β, and IFNλ1 mRNA expression (**Fig 6D**). In fact, fold-induction in levels of pro-inflammatory cytokines at early times post infection (4–6 hpi) was similar between CD169$^+$ and ACE2$^+$ macrophages (**Fig 6D** and **6E**), indicating that early viral RNA production is the key trigger of innate immune activation. Accelerated kinetics and highest magnitude of induced IFNλ1 expression was observed in SARS-CoV-2 infected CD169$^+$/ACE2$^+$ THP1/PMA macrophages (**Fig 6F**, note difference in scale on y axis), which correlated with greater level of negative sense antigenomic RNA (**Fig 4E**) and highest levels of virus replication in CD169$^+$/ACE2$^+$ THP1/PMA macrophages (**Fig 3E**).

To determine if fusion, entry, and initiation of SARS-CoV-2 replication were also required for induction of innate immune responses in infected human primary macrophages, MDMs from multiple donors were treated with TMPRSS2 inhibitor (camostat), cathepsin inhibitor (E64D), or remdesivir (RDV) and infected with SARS-CoV-2 for 24 hours. Total RNA was harvested and subsequently analyzed by RT-qPCR to determine fold-induction of IL-6, TNFα, IL-1β, and IFNλ1 mRNA expression. Whereas camostat treatment did not suppress viral gRNA or sgRNA expression (**Fig 7A** and **7B**) or induction of IL-6, TNFα, IL-1β, and IFNλ1 expression (**Fig 7C**), treatment with E64D significantly decreased both gRNA and sgRNA expression (**Fig 7A** and **7B**), which correlated with abrogation of IL-6, TNFα, IL-1β, and IFNλ1 mRNA induction in a dose-dependent manner (**Fig 7C**). Furthermore, treatment with RDV, which abrogated SARS-CoV-2 replication in MDMs (**Fig 7A** and **7B**), potently inhibited induction of pro-inflammatory cytokine expression (**Fig 7C**). Kinetic analysis for viral

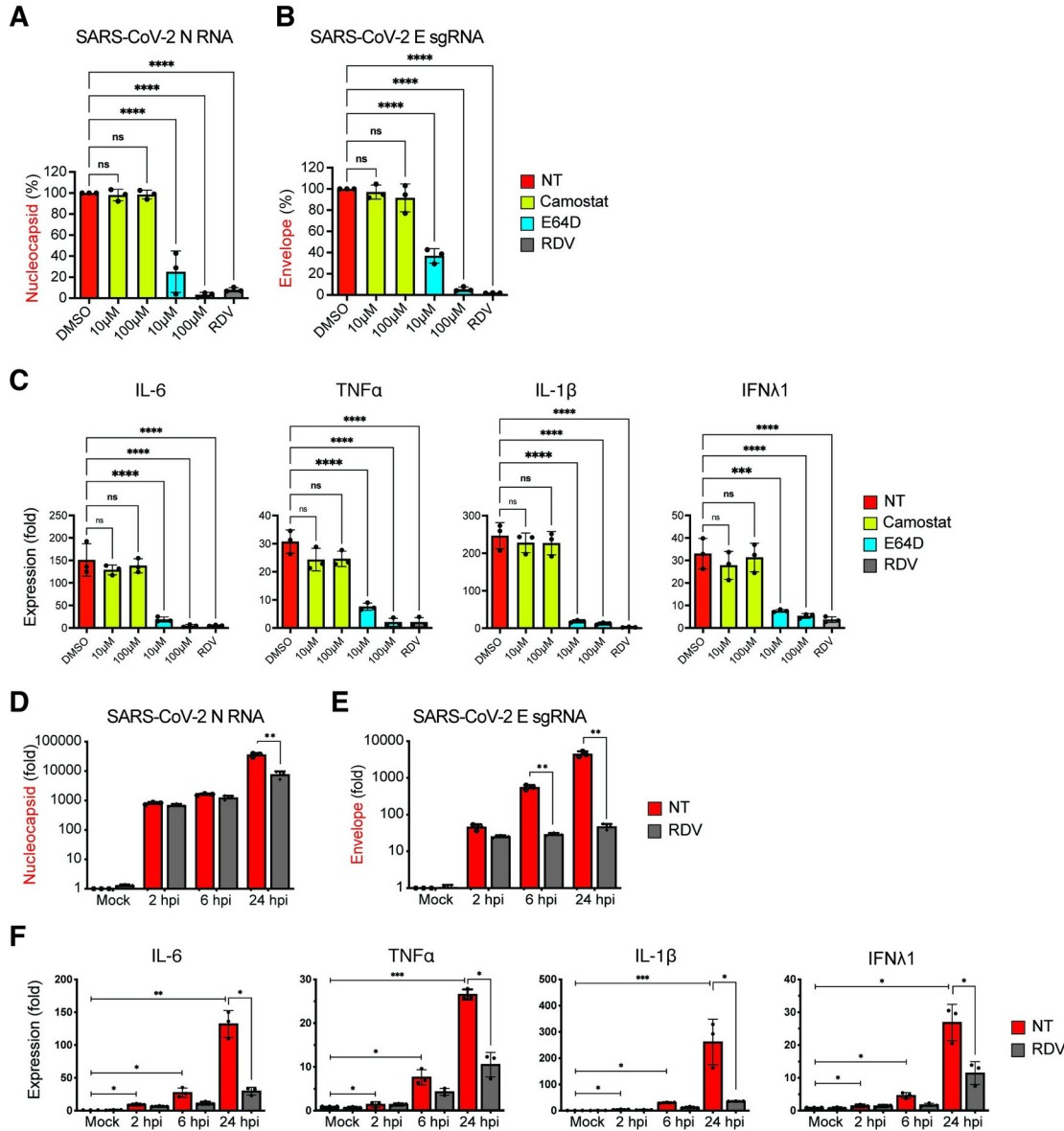

**Fig 7. Endosomal entry of SARS-CoV-2 in MDMs facilitates restricted SARS-CoV-2 RNA synthesis and induction of pro-inflammatory cytokines.** (**A**-**C**) MDMs from multiple donors were infected with SARS-CoV-2 (MOI = 1) in the absence or presence of camostat, E64D, or RDV (1 µM). Cells were harvested at 24 hpi and total RNA analyzed by RT-qPCR for expression of viral N RNA (**A**) and envelope sgRNA (**B**). (**C**) Induction of proinflammatory cytokines normalized to mock (uninfected) control from each donor. (**D**-**F**) MDMs were infected with SARS-CoV-2 (MOI = 1), in the absence or presence of RDV (1 µM), and viral replication kinetics analyzed by RT-qPCR at indicated timepoints for total N RNA (**D**) and E sgRNA (**E**). Induction of indicated cytokine mRNAs (**F**) normalized to mock from each donor. The means ± SEM are shown from 3 independent experiments, and each symbol represents an independent donor. Significant differences between conditions were determined by one-way ANOVA followed by the Dunnett's post-test (**A**-**C**), or Tukey's multiple comparisons test (**F**), comparing to DMSO-treated control. $P$-values: *<0.1; **<0.01; ***<0.001; ****<0.0001; ns: not significant.

transcripts revealed progressive increase in both viral gRNA (**Fig 7D**) and sgRNA (**Fig 7E**) expression up to 24 hpi, levels of which were significantly decreased upon RDV treatment at 24 hpi (**Fig 7D** and **7E**). Similar to the observations in CD169+ THP1/PMA macrophages (**Fig 6B**), viral RNA synthesis (**Fig 7E**) and induction of innate immune responses (**Fig 7F**) in SARS-CoV-2 infected primary MDMs was observed as early as 6 hpi, though induction of pro-

inflammatory cytokine expression was maximal at 24 hpi. Notably, RDV treatment significantly decreased IL-6, TNFα, IL-1β and IFNλ1 expression only at 24 hpi (**Fig 7F**), suggesting that accumulation of viral replication intermediates is required for induction of pro-inflammatory cytokine expression in MDMs. Collectively, these findings suggest that CD169-mediated SARS-CoV-2 entry and endosomal fusion leads to initiation of viral RNA replication and transcription in ACE2-deficient macrophages. Our data further indicate that accumulation of newly synthesized viral RNA might be the trigger for the induction of macrophage-intrinsic inflammatory responses.

## Cytosolic RNA sensing by RIG-I and MDA-5 is required for SARS-CoV-2 induced inflammation in macrophages

Since innate immune activation in THP1/CD169 cells and primary MDMs depends on *de novo* viral RNA synthesis (**Figs 6B** and **7C**), we next sought to delineate the nucleic acid sensing mechanism required for the detection of SARS-CoV-2 gRNA and sgRNAs in CD169[+] THP1/PMA macrophages. Depending on the specific pathogen-derived cytosolic nucleic acids, numerous host sensors can detect and trigger innate immune activation via the MAVS and/or STING pathways [67]. Viral RNAs can be sensed by RIG-I-like receptors (RLRs) or endosomal toll-like receptors (TLRs) to activate MAVS or TRIF, respectively, leading to the induction of pro-inflammatory cytokines [68]. Innate immune sensing of SARS-CoV-2 RNAs by RLRs such as RIG-I and MDA-5, or endosomal TLRs (TLR7/8), has been previously proposed [69,70]. It has also been hypothesized that SARS-CoV-2-induced mitochondrial damage and release of mitochondrial DNA into the cytosol, a cellular stress response, could trigger cGAS/STING sensing pathway in infected cells [71,72]. To investigate which of the nucleic acid sensing pathways are involved in sensing abortive SARS-CoV-2 infection in macrophages, we stably knocked-down expression of RIG-I, MDA-5, UNC93B1 (an adaptor protein required for TLR3/7/9 trafficking to endosomes, [73]), MAVS or STING in CD169[+] THP1/PMA cells through shRNA-based lentiviral transduction. Upon successful selection we confirmed knockdown by showing decrease of both mRNA (**Fig 8A**) and protein expression (**Fig 8B**). Functional validation of knockdowns was performed by a type I IFN bioassay (**S6A Fig**) and IP-10 ELISA (**S6B Fig**). Knock-down of RIG-I, MDA-5, UNC93B1, MAVS or STING in THP1/CD169 macrophages did not impact infection efficiency of SARS-CoV-2, as shown by RT-qPCR analysis of viral gRNA and sgRNAs in each cell line (**Fig 8C** and **8D**). While knock-down of UNC93B1 in virus-infected THP1/PMA CD169[+] cells had negligible impact on induction of pro-inflammatory cytokines (IL-6, TNFα, IL-1β and IFNλ1) compared to scramble control cells, depletion of either RIG-I or MDA-5 led to dramatic reduction in pro-inflammatory cytokine expression to near background levels, suggesting both cytosolic viral RNA sensors, but not endosomal TLRs, are required for innate immune sensing of SARS-CoV-2 transcripts (**Fig 8E**-**8H**). Furthermore, knock-down of MAVS, but not STING, completely abrogated SARS-CoV-2-induced innate immune activation in THP1/PMA CD169[+] macrophages, further confirming the requirement of cytosolic viral RNA sensing for induction of pro-inflammatory cytokines.

## MAVS is essential for SARS-CoV-2 RNA-induced innate immune activation in both non-productively and productively infected THP1/PMA macrophages

We next sought to determine if MAVS was required for the induction for pro-inflammatory responses not only in CD169-mediated abortive infection but also in ACE2-mediated productive infection of macrophages with SARS-CoV-2. Parental THP1 cells or those expressing

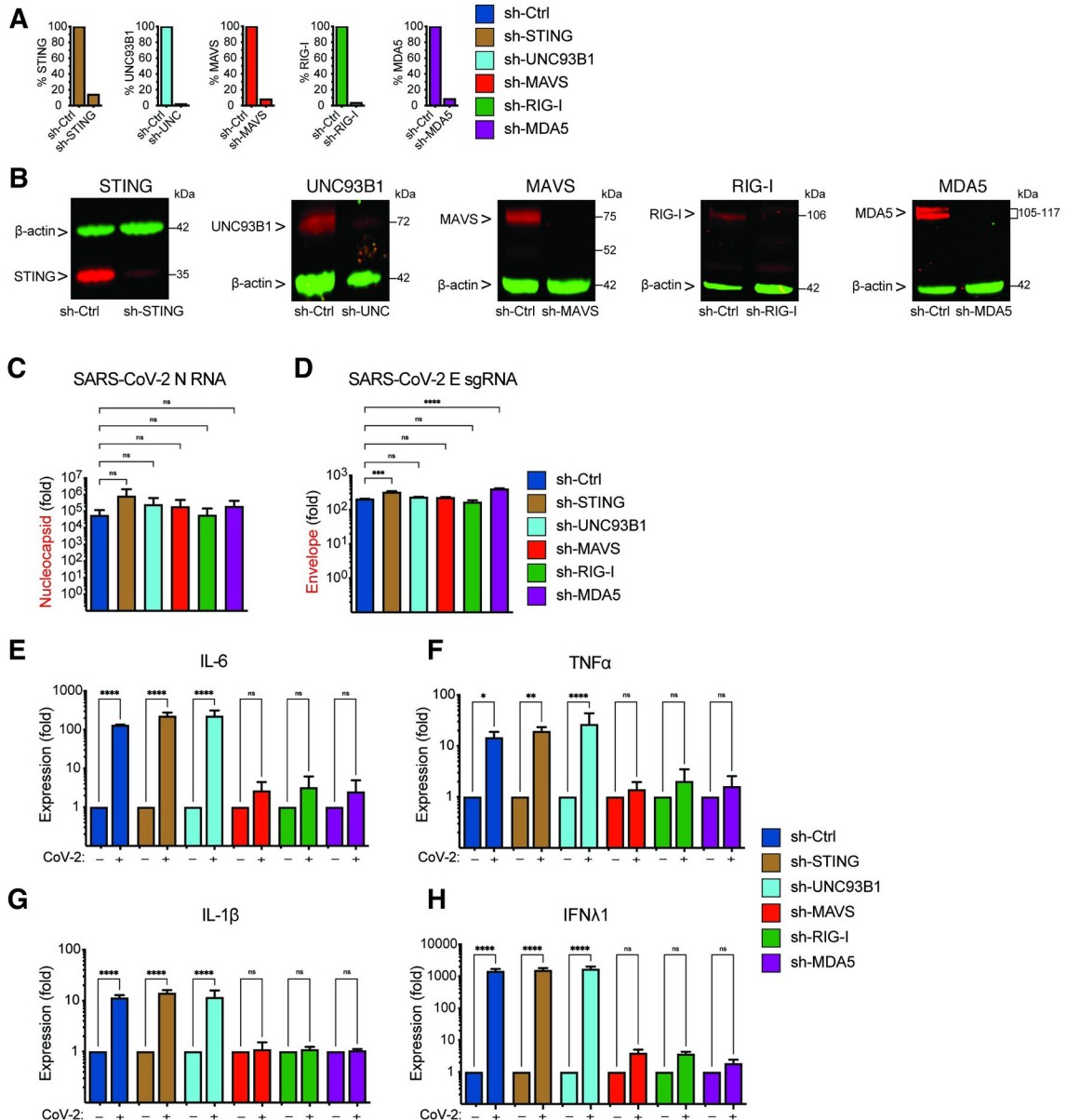

**Fig 8. Viral RNA sensing by RIG-I and MDA5 is required for SARS-CoV-2 induced innate immune responses in macrophages.**
(**A-B**) CD169-expressing THP1 monocytes were transduced with lentiviral vectors expressing shRNA against non-specific control (scramble), or specific shRNA sequences against STING, UNC93B1, MAVS, RIG-I, and MDA5. Knockdown of host proteins targeted by shRNAs compared to scramble analyzed by RT-qPCR (**A**), and immune blotting (**B**). (**C-H**) Total RNA from CD169[+] THP1/PMA macrophages infected with SARS-CoV-2 (MOI = 1, 24 hpi) was analyzed by RT-qPCR for nucleocapsid (gRNA) (**C**), envelope (sgRNA, **D**), and pro-inflammatory cytokines/ISGs, IL6, TNFα, IL-1β and IFNλ1 (**E-F**), normalized to mock-infected controls for each group. Each knockdown is represented by different colors as in **A**. The means ± SEM from 3 independent experiments are shown, and significant differences were determined by one-way ANOVA followed by the Dunnett's post-test comparing to scramble THP1 (**C-D**), or by two-way ANOVA followed by Bonferroni post-test comparing Mock to SARS-CoV-2 infected in each group (**E-H**). *P*-values: *<0.1; **<0.01; ***<0.001; ****<0.0001; ns: not significant.

CD169, ACE2, or both CD169 and ACE2 were transduced with lentiviral vectors expressing scramble shRNAs (control) or MAVS-specific shRNAs, which led to robust knock-down of MAVS protein expression in all cell lines (**Fig 9A**). While MAVS depletion did not affect subsequent SARS-CoV-2 infection of CD169[+], ACE2[+] or CD169[+]/ACE2[+] THP1/PMA cells, as

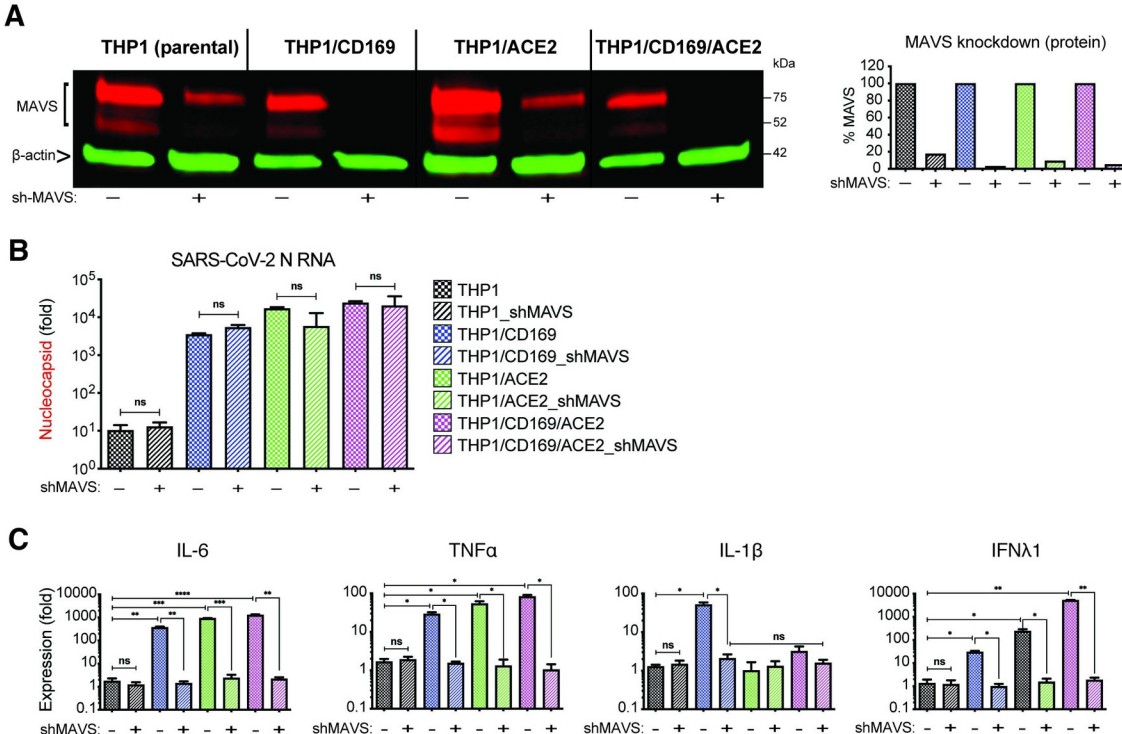

**Fig 9. MAVS is essential for SARS-CoV-2 RNA-induced inflammatory responses in macrophages.** (A) Parental, CD169+, ACE2+, or CD169+/ACE2+ THP1 monocytes were transduced with lentiviral vectors expressing shRNA against scrambled sequence (-), or MAVS sequence (+), and knockdown of MAVS in each cell line was confirmed by western blotting (A). (B-C) MAVS-deficient THP1/PMA macrophages were infected with SARS-CoV-2 (MOI = 1, 24 hpi), and total RNA analyzed by RT-qPCR for (B) viral transcripts (nucleocapsid), and (C) IL6, TNFα, IL-1β and IFNλ1 mRNA expression, normalized to mock-infected controls for each group. The means ± SEM from 3 independent experiments are shown, and significant differences were determined by one-way ANOVA followed by the Tukey-Kramer post-test within groups (**B**, and **C**), or Dunnett's post-test between groups (**C**). *P*-values: *<0.05; **<0.01; ***<0.001; ****<0.0001; ns: not significant.

quantified by total SARS-CoV-2 N gRNA transcripts (**Fig 9B**), induction of pro-inflammatory cytokines (IL-6, TNFα, IL-1β and IFNλ1) was significantly attenuated in both productively (ACE2+, CD169+/ACE2+) and abortively (CD169+) infected THP1/PMA macrophages upon MAVS knockdown (**Fig 9C**). These results suggest that MAVS plays a pivotal role in pro-inflammatory cytokine induction in both ACE2-dependent and independent (CD169-mediated) SARS-CoV-2 infection of THP1/PMA macrophages. Taken together, these findings suggest that CD169-dependent establishment of abortive SARS-CoV-2 infection in macrophages triggers RIG-I/MDA-5 mediated sensing of viral gRNA and sgRNAs and MAVS-dependent inflammatory cytokines induction, which might contribute to the dysregulated hyper-immune phenotype of inflammatory macrophages and severity of COVID-19 disease.

## Discussion

Monocytes and macrophages have not been directly implicated in productive SARS-CoV-2 infection. However, several reports have provided evidence of SARS-CoV-2 RNA and antigen in circulating monocytes, macrophages, and tissue-resident alveolar macrophages, although these cells are not known to express ACE2 (**Fig 1** and [30]). Importantly, a macrophage-intrinsic inflammatory phenotype in lung BALF samples has been associated with COVID-19 disease severity [7]. In this study, we examined the mechanisms by which macrophages potentially contribute to inflammation during SARS-CoV-2 infection, using THP1/PMA and

primary human macrophages expressing a myeloid cell-specific receptor, CD169, in the presence or absence of ACE2.

We showed that CD169 expression mediated entry and fusion of SARS-CoV-2 S-pseudotyped lentivirus in both CD169[+] THP1/PMA and primary macrophages. Other studies, including ours, have also described glycan-dependent interactions of SARS-CoV-2 S protein with C-type lectin receptors (DC-SIGN, LSIGN) and Tweety family member 2 (TTYH2) [12,14,74]. Interestingly, a recent study suggests that CD169 also recognizes gangliosides, such as GM1 and GM3, incorporated in SARS-CoV-2 particle membrane, and can mediate trans infection of ACE2[+] cells [14], similar to our previously reported findings on GM3-dependent CD169-mediated trans infection of HIV-1 [39,43]. However, we show that recombinant S protein also binds to CD169 and that binding is abrogated when the sialic acid binding motif in CD169 is mutagenized (R116A mutant, **Fig 2A**). Furthermore, entry of S pseudotyped lentiviruses in MDMs was attenuated upon pretreatment with either antibodies targeting SARS-CoV-2 S NTD (**S2 Fig**) or endosomal cathepsin inhibitors (**Fig 1D**), suggesting that CD169 capture of SARS-CoV-2 results in an endosomal virus entry mechanism distinct to that mediated by ACE2. This is in agreement with recent data suggesting that binding epitopes on S for myeloid cell-specific receptors are located outside the ACE2 binding domain [12]. Thus, while ACE2-independent mechanisms of SARS-CoV-2 interactions with myeloid cells can be mediated by diverse virus particle-associated antigens (gangliosides, mannosylated S and sialylated S), our results suggest that CD169-S interaction is sufficient to promote virus particle fusion and entry into macrophages in an ACE2-independent manner.

In addition to lectin receptors, recent studies have also described FcγR-mediated entry of antibody-opsonized SARS-CoV-2 in peripheral blood human monocytes [10] and tissue-resident macrophages in humanized mice transduced to express human ACE2 [15]. While infection efficiency in vitro was low (~3% of SARS-CoV-2-exposed monocytes) and only observed upon LPS stimulation [10], infection was enhanced upon exogenous ACE2 expression in tissue-resident macrophages in humanized mice [15]. Similar to these findings, exogenous expression of ACE2 in THP1/PMA macrophages (**Fig 3**) or MDMs (**Fig 5**), rescued virus replication and resulted in production of infectious virus progeny. In contrast, in the absence of ACE2, infectious virus production was not observed in both CD169[+] THP1/PMA macrophages and MDMs. Importantly, induction of inflammatory responses in both productively infected ACE2[+] macrophages and non-productively infected CD169[+] macrophages required initiation of viral replication as treatment with RDV (as described in **Figs 6** and **7**, and in recently published studies ([10,15]) attenuated induction of cytokine expression. Collectively, these findings suggest that SARS-CoV-2 uses diverse entry mechanisms to infect macrophages, leading to a virus-induced pro-inflammatory phenotype.

Pre-treatment with antibodies targeting SARS-CoV-2 S NTD and inhibition of endosomal cathepsins markedly attenuated S-pseudotyped lentiviral infection of CD169[+] macrophages, suggesting that CD169-mediated S-dependent viral entry mechanisms are distinct to those mediated by ACE2. This is in agreement with recent data suggesting that binding epitopes on S for myeloid cell-specific receptors are located outside the ACE2 binding domain [12]. Thus, the results presented here have implications for antibody treatment in COVID-19 patients, as current therapies are primarily focused on antibodies that bind to the ACE2 receptor binding domain (RBD) [57,75–78]. Targeting non-RBD epitopes to disrupt ACE2-independent endosomal entry in CD169[+] myeloid cells could serve as a potential mechanism for preventing myeloid cell-intrinsic immune activation. Indeed, broadly neutralizing nanobodies and potent NTD-targeting neutralizing antibodies from COVID-19 patients that block both ACE2-dependent and ACE2-independent entry were recently described [57,75–78] and might provide potential benefit in suppressing the macrophage-intrinsic inflammatory signature.

SARS-CoV-2 RNA synthesis occurs within ER-derived double-membrane vesicles (DMVs). The establishment of viral replication factories within DMVs in the cytoplasm of infected cells is induced by viral proteins, in concert with cellular factors [64]. Previous reports have utilized dsRNA staining to visualize these viral replication organelles [79]. While viral dsRNA+ cells were observed in a small percentage of THP1/CD169+ macrophages at 6 hpi (**Fig 3B**), no further increase was observed. In contrast, the majority of virus-exposed THP1/ACE2 and THP1/CD169/ACE2 macrophages were dsRNA+ by 24 hpi (**Fig 3C** and **3D**). The paucity of dsRNA positivity in SARS-CoV-2 infected CD169+ macrophages might reflect selective impairment of DMV formation. However, expression of ACE2 in THP-1/PMA and primary MDMs restored infectious virus particle production, suggesting that macrophages are permissive to SARS-CoV-2 replication when entry is facilitated by ACE2. Considering that both CD169 and ACE2 mediated virus entry resulted in similar levels of SARS-CoV-2 gRNA at 6 h pi but only ACE2-mediated virus entry resulted in productive virus infection, these results implicate a hitherto unappreciated post-entry role for ACE2 in virus life cycle in macrophages. Interestingly, expression of both CD169 and ACE2 in macrophages led to enhanced kinetics and magnitude of infection, reflecting an entry-enhancing effect of CD169 even in the context of ACE2-mediated infection, though the mechanism of enhanced kinetics of SARS-CoV-2 replication in CD169+/ACE2+ macrophages remains unclear.

Despite lack of productive infection, cytoplasmic viral RNA expression in CD169+ macrophages potently induced expression of pro-inflammatory cytokines and chemokines. Thus, CD169-mediated viral entry does not simply enable viral uptake by macrophages but also initiates SARS-CoV-2 replication and triggers inflammatory cytokine expression. Critically, treatment with E64D (cathepsin inhibitor) or RDV not only blocked *de novo* viral RNA expression but also significantly reduced pro-inflammatory cytokine expression in non-productively infected CD169+ macrophages, suggesting that viral RNA (replication intermediates) sensing contributed predominantly to inflammatory responses (**Figs 6** and 7). The lack of complete suppression of proinflammatory cytokines upon RDV treatment suggests that virus binding to TLR2 or capture by C-type lectin receptors could also partially induce inflammatory cytokine gene expression [11,12,14,34,35].

Previous reports have implicated both RIG-I and MDA-5 in cytosolic sensing of SARS-CoV-2 RNAs [80,81], although the primary RNA sensor might be cell-type dependent. For instance, recent studies have implicated MDA-5 as the primary viral RNA sensor in lung epithelial cells [82], while other studies have found both MDA-5 and RIG-I sense SARS-CoV-2 infection in Calu-3 cells [81,83]. Our results implicate both RIG-I and MDA-5 in SARS-CoV-2 RNA sensing in macrophages, as knockdown of either RIG-I or MDA-5 significantly attenuated inflammatory cytokine induction in CD169+ macrophages. Despite previous studies suggesting that induction of IL-6 and TNFα might be MAVS-independent [84], knock-down of MAVS abrogated viral RNA sensing and induction of NF-κB-dependent inflammatory cytokines (IL-6, TNFα, IL-1β, IL-18) in productively infected (ACE2+) and abortively infected (CD169+) macrophages. Thus, we propose a co-sensing requirement of both RIG-I and MDA-5 for detecting viral replication intermediates and a pivotal role of MAVS in signal transduction in non-productively infected CD169+ macrophages.

Interestingly, we observed distinct transcriptional signatures for type I and III IFN genes, and proinflammatory cytokines in the different cell lines. While mRNA expression of type I and III IFNs correlated with the extent of viral replication and was highest in THP1/CD169+/ACE2+ macrophages (marked by high levels of IFNλ1 mRNA expression at 24 hpi, **Fig 6A**), the pro-inflammatory cytokines, IL-6 and TNFα, were induced to comparable levels in both productively (ACE2+ and CD169+/ACE2+) and non-productively (CD169+) infected macrophages (**Fig 6A**). In contrast, significant induction of IL-1β and IL-18, was only observed in

THP1/PMA CD169$^+$ macrophages and primary MDMs (**Figs 6** and **7**), suggesting that CD169-mediated viral infection uncouples induction of inflammatory responses from robust viral replication. Intriguingly, induction of RIG-I/MDA-5/MAVS-dependent IFNβ and ISG expression was muted compared to the robust upregulation of NF-kB-dependent pro-inflammatory cytokines in SARS-CoV-2 infected CD169$^+$ macrophages. Since RLR relocalization to the mitochondrial, peroxisomal or ER membranes is thought to initiate robust MAVS-dependent IRF3 activation [85–87], it is tempting to speculate that during restricted SARS-CoV-2 infection and diminished expression of viral RNAs, cytosolic retention or altered intracellular localization of RLRs might control the strength and specificity of downstream responses and favor NF-kB-dependent proinflammatory cytokine expression. Future studies will need to address spatiotemporal dynamics of RIG-I/MDA-5/MAVS interactions upon SARS-CoV-2 RNA sensing in macrophages. Besides MAVS-mediated upregulation of IL-1β and IL-18 expression, viral RNA sensing pathways have also been linked to NLRP3 inflammasome priming and activation via MAVS-mediated recruitment of NLRP3 inflammasome at the mitochondrial membrane for optimal inflammasome activation [88]. Thus, SARS-CoV-2 infected and primed CD169$^+$ macrophages might uniquely contribute to the inflammasome activation upon delivery of secondary activation signals and perpetuate the hyper-inflammatory phenotype.

SARS-CoV-2 RNA infection of lung epithelial cells can contribute to innate immune activation, inflammation, recruitment of inflammatory monocytes and macrophages to the alveolar space, and activation of tissue-resident alveolar macrophages [80,81]. Alveolar macrophages and airway epithelial cell-associated macrophages, which are uniquely positioned as gatekeepers to intercept invading pathogens from the bronchiolar airways, constitutively express CD169 [58,89], an ISG, whose expression can be further induced by type I and type III IFNs [39,50,51,90]. While therapeutic use of both type I and III IFNs have been proposed, clinical benefit has proven inconclusive [91–93], presumably related to timing of the IFN response. For instance, a delayed and persistent type I IFN response without resolution has been correlated with disease severity and mortality in patients with COVID-19 [68]. As was recently suggested [94], the pathological outcomes associated with a delayed type I IFN response might be partly due to the type I IFN-induced expression of SARS-CoV-2 entry receptors, such as CD169, in monocytes/macrophages and increased virus uptake and amplification of inflammatory responses.

Treatment of COVID-19 patients with remdesivir has been shown to significantly shorten recovery time and reduce lower respiratory tract infections, despite having minimal impact on viremia [95,96]. Notably, combination therapy with Baricitinib, an anti-inflammatory drug led to further improvement in clinical status and significantly lowered serious adverse events [97]. Therefore, targeting macrophage-intrinsic innate immune activation by either blocking macrophage-specific receptors which can mediate SARS-CoV-2 uptake, or through administration of RIG-I/MDA-5 antagonists present a novel therapeutic strategy for combating hyperinflammation in COVID-19 patients. However, further research is needed to better understand the mechanisms of ACE2-independent CD169-mediated SARS-CoV-2 infection of macrophages, the post-entry restrictions to virus replication, and the viral determinants that trigger innate immune activation.

## Materials & methods

### Ethics statement

This research has been determined to be exempt by the Institutional Review Board of the Boston University Medical Center since it does not meet the definition of human subjects research, since all human samples were collected in an anonymous fashion and no identifiable private information was collected.

## Plasmids

The SARS-CoV-2 S/gp41 expression plasmid (Wuhan isolate) was a gift from Dr. Nir Hachoen at the Broad Institute, and has previously been described [74]. The pTwist-SARS-CoV-2 Δ18 B.1.617.2v1 (delta) and pTwist-SARS-CoV-2 Δ18 B.1.1.529 (omicron) plasmids (Addgene, Cat# 179905 and 179907) [98] and encode the delta and omicron spike protein lacking 18 C-terminal amino acids. For ACE2 lentiviral transduction in THP1 cells, we used a lentiviral plasmid expressing ACE2 and the puromycin resistant gene (Addgene #145839, hereafter pLenti-ACE2-IRES-puro) [83]. Generation of wildtype and mutant (R116A) human CD169 plasmids (LNC-CD169) was previously described and validated [39]. For transduction of primary MDMs, a 3' LTR-restored lentiviral expression vector (Addgene #101337, hereafter LV-3'LTR) [99] expressing a GFP reporter was used to express ACE2 or CD169. For ACE2 cloning, the NotI-XhoI fragment from pLenti-ACE2-IRES-puro was inserted into the LV-3'LTR backbone. For cloning CD169 into LV-3'LTR vector, a BglII-AgeI fragment from LNC-CD169 was inserted into LV-3'LTR vector. HIV-1 packaging plasmid psPAX2 and VSV-G expression constructs have been previously described [39]. All lentiviral vectors (pLKO.1) expressing shRNAs used for knockdown of host proteins were purchased from Sigma.

## Cells

HEK293T cells (ATCC) were maintained in DMEM (Gibco) containing 10% heat-inactivated fetal bovine serum (FBS) (Gibco) and 1% pen/strep (Gibco) [39,40,100]. Vero E6 cells (ATCC CRL-1586) were maintained in DMEM supplemented with 10% FBS and 100 μg/ml primocin. THP1 cells (ATCC) were maintained in RPMI/1640 (Gibco) containing 10% FBS and 1% pen/strep [50]. THP1 cells stably expressing CD169 have previously been described [39]. To generate HEK293T/ACE2[+], HEK293T/CD169, THP1/ACE2[+] and THP1/CD169[+]/ACE2[+] cells, HEK293T, THP1 or THP1/CD169 cells were transduced with Lenti-ACE2-IRES-puro lentivector or LNC-CD169 retroviral vector and cultured in puromycin (2 μg/ml) or G418 (1 mg/ml) containing media. Cells with robust surface expression of ACE2, CD169 and CD169/ACE2 (double-positive) were sorted using a MoFlo cell sorter (Beckman Coulter) and cultured in puromycin or G418-supplemented media. All cell lines are routinely tested for mycoplasma contamination and confirmed negative. For THP1 monocyte to macrophage differentiation, THP1 cells were stimulated with 100 nM PMA (Sigma-Aldrich) for 48 hours. Human monocyte-derived macrophages (MDMs) were derived from CD14[+] peripheral blood monocytes by culturing cells in RPMI/1640 (Gibco) media containing 10% heat-inactivated human AB serum (Sigma-Aldrich) and recombinant human M-CSF (20 ng/ml; Peprotech) for 5–6 days, [55]. To generate MDMs expressing ACE2 or overexpressing CD169, cells were co-infected with ACE2 or CD169 expressing lentiviruses (100 ng based on p24$^{gag}$ ELISA per 1x10$^6$ cells) and SIV3+ (Vpx expressing) VLPs. ACE2 and CD169 surface expression was determined by flow cytometry 3 days post transduction.

## Viruses

SARS-CoV-2 stocks (isolate USA_WA1/2020, kindly provided by CDC's Principal Investigator Natalie Thornburg and the World Reference Center for Emerging Viruses and Arboviruses (WRCEVA)) and recombinant SARS-CoV-2 expressing mNeonGreen (SARS-CoV-2-mNG, kindly provided by Pey-Yong Shi, University of Texas Medical Branch, Galveston and the World Reference Center for Emerging Viruses and Arboviruses [101], based on SARS-CoV-2 isolate USA_WA1/2020) were grown in Vero E6 cells (ATCC CRL-1586) cultured in Dulbecco's modified Eagle's medium (DMEM) supplemented with 2% fetal FBS and 100 μg/ml primocin. To remove confounding cytokines and other factors, viral stocks were purified by

ultracentrifugation through a 20% sucrose cushion at 80,000xg for 2 hours at 4˚C [60]. SARS-CoV-2 titer was determined in Vero E6 cells by tissue culture infectious dose 50 (TCID$_{50}$) assay using the Spearman Kärber algorithm. All work with SARS-CoV-2 was performed in the biosafety level 4 (BSL4) facility of the National Emerging Infectious Diseases Laboratories at Boston University, Boston, MA following approved SOPs.

Generation of SARS-CoV-2 S-pseudotyped lentiviruses expressing spike glycoprotein has previously been described [74]. Briefly, HEK293T cells were co-transfected with HIV-1 reporter plasmid containing a luciferase reporter gene in place of the *nef* ORF and SARS-CoV-2 S [74]. To generate ACE2 or CD169 expressing recombinant lentiviruses, LV-3'LTR lentivectors expressing either empty vector, ACE2, or CD169, were co-transfected with psPax2 (HIV Gag-pol packaging plasmid) and H-CMV-G (VSV-G envelope-expressing plasmid) in HEK293T cells by calcium phosphate-mediated transfection [102]. Virus-containing supernatants were harvested 2 days post-transfection, cleared of cell debris by centrifugation (300xg, 5 min), passed through 0.45 μm filters, aliquoted and stored at -80˚C until use. Lentivirus titers were determined by a p24$^{gag}$ ELISA [102].

## Infection

For RNA analysis, 1x10$^6$ cells (THP1/PMA, MDMs, HEK293T) were seeded per well of a 12-well plate. For smFISH and immunofluorescence analysis, 1x10$^6$ cells (THP1/PMA or MDMs) were seeded per well of a 6-well plate containing 3–4 coverslips per well, or 1x10$^4$ cells were seeded per well of a 96-well plate (HEK293T). The next day, cells were infected with purified SARS-CoV-2 or SARS-CoV-2-mNG stocks at the indicated multiplicity of infection (MOI). After a 1-hour adsorption period at 37˚C, virus inoculum was removed, the cells were washed twice with PBS, and fresh cell culture medium was added. For drug treatments (E64D, camostat, remdesivir) the cells were pre-treated for 30 minutes at 37˚C prior to infection and virus was added in the presence of drugs. Virus was removed, the cells were washed as described above, and fresh cell culture medium containing the respective drug was added. At the indicated time points, the cells were either lysed with TRIzol (for total RNA analysis) or fixed in 10% neutral buffered formalin for at least 6 hours at 4˚C and removed from the BSL-4 laboratory for staining and imaging analysis in accordance with approved SOPs. For SARS-CoV-2 S-pseudotyped lentiviral infections of THP1/PMA macrophages or MDMs, 1x10$^5$ cells were seeded in 96-well plates, and infected via spinoculation the following day with 10–20 ng (p24$^{Gag}$) of purified S-pseudotyped lentivirus and SIV$_{mac}$ Vpx VLPs (1 hr at RT and 1100 x g), as previously described [55]. Incubation with virus was continued for 4 additional hours at 37˚C, cells were then washed to remove unbound virus particles, and cultured for 2–3 days. For CD169 blocking experiments, primary MDMs from 3 different donors were pre-incubated with 20 μg/ml anti-CD169 antibody (HSn 7D2, Novus Biologicals) or IgG1k (P3.6.2.8.1, eBioscience) for 30 min at 4˚C prior to infection. For anti-spike neutralizing experiments, virus-containing media were pre-incubated with antibodies targeting SARS-CoV-2 spike NTD for 30 min at 37˚C prior to infection. At indicated timepoints, cells are lysed, cell lysates were analyzed for luciferase activity using the Bright-Glo luciferase assay kit (Promega), as previously described [36]. The SARS-CoV-2 neutralizing antibodies was previously characterized [57] and were a kind gift from Dr. Duane Wesemann at Harvard Medical School.

## S binding

To evaluate SARS-CoV-2 S binding to various THP1 monocytes expressing different surface receptors, approximately 0.25x10$^6$ cells from parental THP1 or those expressing wt CD169, mutant CD169 (R116A), ACE2, or both wt CD169 and ACE2 were incubated for 30 min at

4˚C with 2 μg of spike glycoprotein (stabilized) from Wuhan-Hu-1 SARS-CoV-2 containing a C-terminal Histidine Tag, recombinant from HEK293F cells (BEI resources, #NR-52397). This is followed by secondary staining for 30 min at 4˚C with APC-conjugated mouse anti-His antibody (BioLegend, #362605, 1:50) or isotype control. Cells were fixed with 4% PFA (Boston Bioproducts) for 30 min, and analyzed with BD LSRII (BD). Data analysis was performed using FlowJo software (FlowJo).

## Immunofluorescence

In brief, the cells were permeabilized with acetone-methanol solution (1:1) for 10 min at -20˚C, incubated in 0.1 M glycine for 10 min at room temperature and subsequently incubated in 5% goat serum (Jackson ImmunoResearch) for 20 minutes at room temperature. After each step, the cells were washed three times in PBS. The cells were incubated overnight at 4˚C with a rabbit antibody directed against the SARS-CoV nucleocapsid protein (Rockland; 1:1000 dilution in 5% goat serum), which cross-reacts with the SARS-CoV-2 nucleocapsid protein, as previously described [103]. The cells were washed four times in PBS and incubated with goat anti-rabbit antibody conjugated with AlexaFluor594 for 1 hour at room temperature (Invitrogen; 1:200 dilution in blocking reagent). 4',6-diamidino-2-phenylindole (DAPI; Sigma-Aldrich) was used at 200 ng/ml for nuclei staining. For dsRNA staining [61], anti-dsRNA (Pan-Enterovirus Reagent, clone 9D5, Light Diagnostics, Millipore) antibody was used 1:2 overnight and anti-mouse-AF488 (Invitrogen) 1:200 dilution as secondary antibody with DAPI. Images were acquired using a Nikon Eclipse Ti2 microscope with Photometrics Prime BSI camera and NIS Elements AR software.

## RNA isolation and RT-qPCR

Total RNA was isolated from infected cells using TRIzol reagent (Invitrogen). Reverse transcription (RT) from purified RNAs was performed using oligo(dT)$_{20}$ primer (Superscript III, Invitrogen) or strand-specific RT primers as previously described [104]. Briefly, to detect negative-strand viral RNA, the RT primer,

5'- **ACAGCACCCTAGCTTGGTAG**CCGAACAACTGGACTTTATTGA -3' was used, and it consists of two parts. The first sequence is labeled in bold and underlined, and serves as an internal amplification control which is not present in the viral RNAs. The second part targets ORF1ab gene from 647 to 688bp of SARS-CoV-2 (isolate USA_WA1/2020). For amplification, the forward primer, 5'- AGGTGTCTGCAATTCATAGC-3' (743-762bp), and the reverse primer, 5'- **ACAGCACCCTAGCTTGGTAG** -3' were used. As a negative control for amplification, the primer, 5'-CCCTGATGGCTACCCTCTT-3' (583-602bp) was used. RT was performed using Superscript III 1st Strand cDNA Synthesis (Invitrogen), and PCR was carried out using Maxima SYBR Green (Thermo Scientific). Viral RNAs and host mRNAs were quantified using Maxima SYBR Green (Thermo Scientific), using the primer sets shown in **Table 1**. Primer sequences for GAPDH, IL-6, TNFα, IL-1β, IP-10, IFNλ1, IL-18, Viperin and IFNβ have been described previously [100]. The $C_T$ value was normalized to that of GAPDH and represented as a relative value to a 'mock' control using the $2^{-\Delta\Delta C_T}$ method as described [100,105].

## shRNA mediated knockdown

Stable knockdown of host proteins in THP1 cells was carried out by transduction with lentivectors expressing individual shRNAs (pLKO.1, 400 ng p24$^{Gag}$ (as measured by ELISA) per $1 \times 10^6$ cells) in the presence of polybrene (Millipore). All shRNA target sequences are listed in **Table 2**. Cells were washed and cultured for 5–7 days in the presence of puromycin (2 μg/ml,

**Table 1. qRT-PCR primers.**

| Gene | Forward | Reverse |
|---|---|---|
| GAPDH | CAAGATCATCAGCAATGCCT | AGGGATGATGTTCTGGAGAG |
| IL-6 | TCTCCACAAGCGCCTTCG | CTCAGGGCTGAGATGCCG |
| TNFα | CCCAGGGACCTCTCTCTAATCA | GCTACAGGCTTGTCACTCGG |
| IL-1β | AAACAGATGAAGTGCTCCTTCC | AAGATGAAGGGAAAGAAGGTGC |
| IL-18 | GACCAAGGAAATCGGCCTCTA | ACCTCTAGGCTGGCTATCTTTATACATAC |
| IFNβ | ATTCTAACTGCAACCTTTCG | GTTGTAGCTCATGGAAAGAG |
| IP-10 | AAAGCAGTTAGCAAGGAAAG | TCATTGGTCACCTTTTAGTG |
| Viperin | TGGGTGCTTACACCTGCTG | GAAGTGATAGTTGACGCTGGTT |
| IFNλ1 | GGACGCCTTGGAAGAGTCAC | AGCTGGGAGAGGATGTGGT |
| SARS-CoV-2_ negative strand | ACAGCACCCTAGCTTGGTAGCCGAACAACTGGACTTTATTGA | |
| nCoVsg_E.rtF | CGAACTTATGTACTCATTCGTTTCGG | |
| nCoVsg_E.rtR | AGAAGGTTTTACAAGACTCACGTT | |
| SARS-CoV-2 Nucleocapsid | CACATTGGCACCCGCAATC | GAGGAACGAGAAGAGGCTTG |
| SARS-CoV-2 Envelope (E) | ACAGGTACGTTAATAGTTAATAGCGT | ATATTGCAGCAGTACGCACACA |
| TRS 5' leader (forward) | ACCAACCAACTTTCGATCTCTTGT | |
| N 3' reverse | CACTGCGTTCTCCATTCTGG | |
| ACE2 | CGAAGCCGAAGGCCTGTTCTA | GGGCAAGTGTGGACTGTTCC |
| TMPRSS2 | CAAGTGCTCCAACTCTGGGAT | AACACACCGATTCTCGTCCTC |
| STING | ACTGTGGGGTGCCTGATAAC | TGGCAAACAAAGTCTGCAAG |
| MAVS | GTACCCGAGTCTCGTTTC | GCAGAATCTCTACAACATCC |
| RIG-I | ATCCCAGTGTATGAACAGCAG | GCCTGTAACTCTATACCCATGTC |
| MDA-5 | GGCATGGAGAATAACTCATCAG | CTCTTCATCTGAATCACTTCCC |
| UNC93B1 | TGATCCTGCACTACGACGAG | GCGAGGAACATCATCCACTT |

InvivoGen). Selected cells were expanded, and knockdown confirmed and quantified by RT-qPCR or western blotting, prior to any downstream experiments. For functional analysis of all knockdown cell lines, THP1 monocytes from each line were stimulated with respective ligands for 24 hours, and conditioned media harvested for type I IFN bioassay [100] and human IP-10 ELISA (BD). The following RLR or TLR ligands were used for the functional validation: cGAMP (Invivogen #tlrI-nacga23), 5 μg/ml, imiquimod (Novus Bio #NBP2-26228), 1 μg/ml, 3'hp-RNA (Invivogen #tlrI-hprna), 2.5 ng/ml, and HMW poly I:C (Invivogen #tlrI-pic), 25 μg/ml.

## Flow cytometry

To examine cell surface expression of CD169 or ACE2 in transduced HEK293T, THP1 or primary MDMs, approximately 0.5x10^6 cells were harvested with CellStripper (Corning), stained with Zombie-NIR (BioLegend, #423105, 1:250) followed by staining for 30 min at 4°C with the

**Table 2. shRNA sequences.**

| Gene | Sequence | Reference |
|---|---|---|
| STING | GCCCGGATTCGAACTTACAAT | Sigma (TRCN0000163296) |
| MAVS | ATGTGGATGTTGTAGAGATTC | Sigma (TRCN0000236031) |
| RIG-I | CCAGAATTATCCCAACCGATA | Sigma (TRCN0000153712) |
| MDA-5 | CCAACAAAGAAGCAGTGTATA | Sigma (TRCN0000050849) |
| UNC93B1 | CAAGGAGAGACAGGACTTCAT | Sigma (TRCN0000138268) |

following antibodies; Alexa647-conjugated mouse anti-CD169 antibody (BioLegend, #346006, 1:50), Alexa647-conjugated mouse anti-ACE2 antibody (R&D systems, 1:200), or unconjugated goat anti-ACE2 polyclonal antibody (R&D systems, #AF933, 1:200) followed by Alexa488-conjugated chicken anti-goat antibody (Invitrogen, #A-21467, 1:100). Cells were fixed with 4% PFA (Boston Bioproducts) for 30 min, and analyzed with BD LSRII (BD). Data analysis was performed using FlowJo software (FlowJo).

## smFISH

**Probe designs.** smFISH probes used to detect different RNA segments of the SARS-CoV-2 genome (NCBI reference sequence: NC_045512.2) consisted of a set of 48 oligonucleotides, each with length of 20 nt and labeled with different fluorophores (see **S1 Table**) for target genes and sequences for each of the probe sets). Probes were designed using Stellaris Probe Designer by LGC Biosearch Technologies and purchased from Biosearch Technologies. The 3'-end of each probe was modified with an amine group and coupled to either tetramethylrhodamine (TMR; Thermo Fisher Scientific), Texas Red-X (Thermo Fisher Scientific), Quasar 670 (Biosearch Technologies) or Cy5 (Lumiprobe). Coupled probes were ethanol precipitated and purified on an HPLC column to isolate oligonucleotides linked to the fluorophore via their amine groups, as previously described by *Raj et.al.* [62].

**Hybridization.** Cells were cultured on glass coverslips, fixed at appropriate times with 10% neutral buffered formalin and permeabilized with 70% methanol. Coverslips were equilibrated with hybridization wash buffer (10% formamide, 2X SSC), and then immersed in 50 μL of hybridization buffer, which consisted of 10% dextran sulphate (Sigma-Aldrich), 1 mg/ml *Escherichia coli* transfer RNA (Sigma-Aldrich), 2 mM ribonucleoside vanadyl complexes (New England Biolabs, Ipswich, MA), 0.02% ribonuclease-free bovine serum albumin (Thermo Fisher Scientific), 10% formamide, 2X SSC, and conjugated probes with appropriate concentration (25 ng of pooled probes). This hybridization reaction mixture was first added as a droplet onto a stretched-out piece of Parafilm (Bemis in North America, Oshkosh, WI) over a glass plate, and then a coverslip containing the cells was placed faced down onto the droplet, followed by incubation at 37˚C overnight in a humid chamber. Following hybridization, the coverslips were washed twice for 10 minutes each in 1 ml of hybridization wash buffer at room temperature. The coverslips were then equilibrated with mounting buffer (2X SCC, 0.4% glucose) and mounted in the mounting buffer supplemented with 1 μL of 3.7 mg/ml glucose oxidase and 1 μl of catalase suspension (both from Sigma-Aldrich) for each 100 μl preparation. After removing the excess mounting medium by gently blotting with a tissue paper, the coverslips were sealed with clear nail polish, and then imaged on the same day.

**Image acquisition, pre-processing, analysis and mRNA quantification.** Images were acquired using Zeiss Axiovert 200M (63x oil immersion objective; numerical aperture 1.4) controlled by Metamorph image acquisition software (Molecular Devices, San Jose, CA). Stacks of images of 16 layers with 0.2 μm interval at 100- to 2,000-milisecond exposure times were used in each fluorescence color channel including DAPI. Two representative coverslips per sample/group were selected and 10–20 regions/fields of interest were imaged. For cell fluorescence intensity measurements, region of interest was drawn manually around each cell using DIC and DAPI channels, then average intensity was measure within the area of each cell using RNA-specific florescence channels.

## Immunoblot Analysis

To assess expression of endogenous or transduced proteins, cell lysates containing 30–40 μg total protein were separated by SDS-PAGE, transferred to nitrocellulose membranes and the

membranes were probed with the following antibodies: mouse anti-TMPRSS2 (Santa Cruz, #515727, 1:1000), mouse anti-Cathepsin-L (Santa Cruz, #32320, 1:1000), goat anti-ACE2 (R&D systems, #AF933, 1;1000), rabbit anti-STING (Cell Signaling, #13647, 1:1000), rabbit anti-MAVS (Thermo Fisher, #PA5-17256, 1:1000), mouse anti-RIG-I (AdipoGen, #20B-0009, 1:1000), rabbit anti-MDA-5 (Proteintech, #21775-1-AP, 1:1000), rabbit anti-UNC93B1 (Invitrogen, #PA5-83437, 1:1000), rabbit anti-IRF1 (Cell Signaling, #8478S, 1:1000). Specific staining was visualized with secondary antibodies, goat anti-mouse-IgG-DyLight 680 (Thermo Scientific, #35518, 1:20000), goat anti-rabbit-IgG-DyLight 800 (Thermo Scientific, #SA5-35571, 1:20000), or a donkey anti-goat-IgG-IR-Dye 800 (Licor, #926–32214, 1:20000). As loading controls, actin or tubulin expression was probed using a rabbit anti-actin (Sigma-Aldrich, A2066, 1:5000), mouse anti-actin (Invitrogen, #AM4302, 1:5000), or rabbit anti-tubulin (Cell Signaling, #3873, 1:5000). Membranes were scanned with an Odessy scanner (Li-Cor).

## Statistics

All the statistical analysis was performed using GraphPad Prism 9. $P$-values were calculated using one-way ANOVA followed by the Tukey-Kramer post-test (symbols for $p$-values shown with a line) or the Dunnett's post-test (comparing to mock), symbols for $p$-values shown with a bracket), or a two-tailed paired t-test (comparing two samples, symbols for two-tailed $p$-values shown with a line bracket). Symbols represent, *: $p < 0.05$, **: $p < 0.01$, ***: $p < 0.001$, ****: $p < 0.0001$. No symbol or ns: not significant ($p \geq 0.05$).

## Supporting information

**S1 Fig. Expression of endogenous TMPRSS2 and Cathepsin-L in human macrophages.** (**A**-**B**) Western blot analysis for TMPRSS2, uncleaved form detected in HEK293T and macrophages (**A**) and the different isoforms of Cathepsin-L, detected in in THP1 and macrophages (**B**) expression in wildtype and transduced HT-29 cells (control), HEK293T, THP1/PMA macrophages, and primary MDMs from multiple donors. β-actin was probed as a loading control. (TIF)

**S2 Fig. Infection of primary MDMs by S-pseudotyped lentivirus is blocked by anti-SARS-CoV-2 NTD antibodies.** SARS-CoV-2 S-pseudotyped lentivirus (20 ng based on p24$^{Gag}$) was pre-incubated with indicated anti-Spike neutralizing antibodies for 30 mins at 37˚C, followed by infection of primary MDMs for 3 days. Relative infection quantified by luciferase activity from whole cell lysates. Data are representative of 2 independent experiments, from 3 different donors each. Mock: no virus added, PBS: no pre-incubation of virus with antibody. The means ± SEM are shown. $P$-values: one-way ANOVA followed by the Dunnett's post-test comparing to untreated (PBS) control, *: $p < 0.05$; **: $p < 0.01$; ***: $p < 0.001$; ****: $p < 0.0001$. (TIF)

**S3 Fig. Expression profiles of human CD169 and ACE2 in THP1 monocytes and primary MDMs.** (**A**) Transduced THP1 cell lines stably expressing wild type (wt) CD169, mutant (R116A) CD169, ACE2, or both wt CD169 and ACE2. (**B**) Untransduced primary MDMs from multiple donors showing differential expression of endogenous CD169. After 5–6 days of macrophage differentiation, cells were either unstained or stained with anti-human CD169 antibody, and surface expression analyzed by flow cytometry. (**C**-**D**) Representative flow cytometry profiles of primary MDMs transduced with wt CD169 (**C**) or ACE2 (**D**) lentiviruses compared to negative (vector only) control. (TIF)

**S4 Fig. CD169 mediated SARS-CoV-2 entry in HEK293T cells fails to rescue productive infection.** (**A**) Transduced or untransduced HEK293T cells were infected with either Wuhan, Delta, or Omicron S-pseudotyped lentivirus (20 ng p24$^{Gag}$) and infection quantified by measuring luciferase expression at 3 dpi. RLUs from each cell line were normalized were normalized to no virus (mock) control. The means ± SEM are shown from 3 independent infections. Significant differences between conditions were determined by one-way ANOVA followed by Tukey's multiple comparisons test, comparing to untransduced HEK293T cells. *P*-values: *<0.1; **<0.01; ***<0.001; ****<0.0001. (**B**) Representative immunofluorescence images (20x) of HEK293T cells overexpressing CD169 or ACE2 compared to untransduced parental line. Cells were treated with DMSO (control) or remdesivir (RDV, 1 μM) for 30 minutes, infected with SARS-CoV-2mNG (MOI = 1) in the absence or presence of RDV, and fixed at 24 hpi followed by staining with DAPI. Images represent 3 independent infections for each cell line. Bar = 25 μm.
(TIF)

**S5 Fig. Absence of pro-inflammatory cytokine expression in THP1/PMA macrophages infected with SARS-CoV-2 S-pseudotyped lentiviruses.** PMA-differentiated THP1 cell lines were infected with SARS-CoV-2 S-pseudotyped lentivirus (20 ng p24$^{Gag}$) and total RNA was harvested at 2 dpi, followed by qRT-PCR analysis. Fold expression of indicated cytokines normalized to mock (uninfected) condition in each group. Data are representative of at least 3 independent experiments. The means ± SEM from 3 independent experiments are shown.
(TIF)

**S6 Fig. Functional analysis of stable knockdown of innate immune sensing pathways in THP1 cells.** Parental or knockdown (STING, UNC93B1, MAV, RIG-I or MDA5) THP1 cells were stimulated for 24–48 hours with ligands targeting the respective innate immune sensor/adaptor for 24–48 hrs, followed by analysis of culture supernatants for type I IFN production by a bioassay (**A**), or IP-10 production by ELISA (**B**). All values normalized to mock (no stimulation) for each cell line.
(TIF)

**S1 Table. smFISH probe sequences.**
(XLSX)

## Acknowledgments

We thank the BUMC Flow Cytometry Core, the Cellular Imaging Core, and Mitchell White, BU for technical assistance. We are grateful to Robert Davey, BU for help with imaging. The following plasmids obtained from Addgene were generous gifts from Dr. Alejandro Balazs (Cat# 179905 and 179907), Dr. Caroline Goujon (Cat#145839), and Dr. Jeremy Luban (Cat#101337).

## Author Contributions

**Conceptualization:** Elke Mühlberger, Suryaram Gummuluru.

**Data curation:** Sallieu Jalloh, Judith Olejnik, Annuurun Nisa, Sita Ramaswamy, Sanjay Tyagi, Yuri Bushkin, Elke Mühlberger, Suryaram Gummuluru.

**Formal analysis:** Sallieu Jalloh, Judith Olejnik, Sanjay Tyagi, Suryaram Gummuluru.

**Funding acquisition:** Sanjay Tyagi, Yuri Bushkin, Elke Mühlberger, Suryaram Gummuluru.

**Investigation:** Judith Olejnik, Jacob Berrigan, Annuurun Nisa, Ellen L. Suder, Hisashi Akiyama, Maohua Lei, Suryaram Gummuluru.

**Methodology:** Sallieu Jalloh, Jacob Berrigan, Hisashi Akiyama, Elke Mühlberger.

**Project administration:** Yuri Bushkin, Suryaram Gummuluru.

**Supervision:** Sanjay Tyagi, Yuri Bushkin, Elke Mühlberger, Suryaram Gummuluru.

**Visualization:** Sanjay Tyagi.

**Writing – original draft:** Sallieu Jalloh, Suryaram Gummuluru.

**Writing – review & editing:** Sallieu Jalloh, Judith Olejnik, Sanjay Tyagi, Yuri Bushkin, Elke Mühlberger, Suryaram Gummuluru.

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
