## [Decision Letter · Decision Letter 0]

16 May 2022

Dear Prof. Gummuluru,

Thank you very much for submitting your manuscript "CD169-mediated restrictive SARS-CoV-2 infection of macrophages induces pro-inflammatory responses" for consideration at PLOS Pathogens. As with all papers reviewed by the journal, your manuscript was reviewed by members of the editorial board and by several independent reviewers. In light of the reviews (below this email), we would like to invite the resubmission of a significantly-revised version that takes into account the reviewers' comments. For resubmission, it would be critical to include substantial experimental evidence in primary cells with infectious SARS-CoV-2 as highlighted by reviewers.

We cannot make any decision about publication until we have seen the revised manuscript and your response to the reviewers' comments. Your revised manuscript is also likely to be sent to reviewers for further evaluation.

Sincerely,

Craig B Wilen, MD, PhD

Reviews Editor

PLOS Pathogens

Michael Diamond

Section Editor

PLOS Pathogens

Kasturi Haldar

Editor-in-Chief

PLOS Pathogens

orcid.org/0000-0001-5065-158X

Michael Malim

Editor-in-Chief

PLOS Pathogens

orcid.org/0000-0002-7699-2064

Reviewer's Responses to Questions

**Part I - Summary**

Reviewer #1: This manuscript by Jalloh and Olejnik et al from the Suyaram and Elke group, describes the interesting role of the I-type lectin CD169/Siglec-1 in promoting SARS-CoV-2 entry into macrophages in an ACE2-independent manner. The authors use parallel studies using THP-1 monocyte cell line model and primary monocyte-derived macrophages (MDMs) to decribes several interesting features of CD169-dependent SARS-CoV-2 infection. Importantly, CD169-mediated SARS-CoV-2 entry is 1. ACE2- and TMPRSS2-independent 2. Cathepsin-dependent 3. results in abortive replication with production of viral genomic and subgenomic RNA but limited expression of viral protein 4. RIG-MDA5-mediated sensing of viral RNA 4. induces IFN, ISGs and IFN lamda 5. induces pro-inflammatory cytokines including those typically triggered by inflammasomes like IL-1b and IL-18. Overall the study is well conducted, interpreted and discussed.

However given the spate of papers on macrophage infection by SARS-CoV-2 (PMID:35385861, 34611663, 33758831, 33277988) the authors will need to interpret/discuss their data in light of these publications/preprints. Importantly, the Lieberman and Flavell group implicate FcgR (not CD169) and opsonization of virus in promoting abortive SARS-CoV-2 entry and induction of inflammasomes. That said, the current work remains very important and occupies a different niche and may serve as a model to understand the role of CD169 during early steps of infection (right after infection of epithelial cells) when SARS-CoV-2 antibodies are not yet circulating in the system and may be FcgR-independent.

Reviewer #2: Jalloh et al. evaluate the ACE2-independent entry of SARS-CoV-2 in human monocyte-derived macrophages (MDM) and determine that CD169 mediates entry but is ultimately restricted. Despite this restriction, the investigators evaluate the upregulation and activation of inflammatory cytokines and interferon-stimulated genes including IL6, TNFa, IL1beta and the impact of inhibition of STING, RIG-1, MAVS, and MDA5 among other viral sensing proteins. Overall, this is an interesting study and sheds new insights on whether SARS-CoV-2 can infect, even if abortively, macrophages via ACE2-independent mechanisms.

Reviewer #3: SARS=CoV-2 infection triggers inflammation within the lungs that is in part driven by immune responses by monocytes and macrophages. Recent studies have found that viral antigen and RNA can be found in macrophages in the absence of any obvious virus replication within these cells. In this manuscript, the authors examine the interaction between SARS-CoV-2 and macrophages, with specific focus on CD169, a c-type lectin. The authors found that SARS-CoV-2 can enter and fuse to macrophages in an ACE2-independent manner by using CD169. These cells did not support productive virus replication (no infectious virus production). Virus infection resulted in a MAVS-dependent pro-inflammatory cytokine response. The relevance of these findings in SARS-CoV-2 pathogenesis are unclear and most of the studies were performed using cultured THP1 cells modified to express CD169 and/or ACE2. Further, there was a strong reliance on using SARS-CoV-2 pseudotyped lentivirus to show the use of CD169 on virus entry and infection of macrophages.

**Part II – Major Issues: Key Experiments Required for Acceptance**

Reviewer #1: 1. Interpetation/discussion of the data wrt to recently published observations is required (PMID:35385861, 34611663, 33758831, 33277988).

2. In my opinion the use of lentivirus pseudotyped Spike virus was an unfortunate choice for this study given the established role of CD169 in promoting lenti/retroviral infections. As lenti/retrovirus capture by CD169 can occur independent of viral env glycoprotein by binding to sialosides on viral membranes, it is difficult to interpret these data with clarity due to the parallel lentivirus biology at play. A more prudent choice for these experiments would have been SARS-CoV-2 VLPs (PMID:34735219) given that SARS-CoV-2 viral membranes are not plasma membrane-derived like lenti/retroviruses but are from ER-Golgi compartments and may have different set of gangliosides. However parallel data with SARS-CoV-2 live virus somewhat obviates this concern. But the authors should consider acknowledging this limitation.

Reviewer #2: 1. In the experiments shown in figure 2, the investigators show that CD169 mediates part of this infection in THP1 monocytic cell lines. Did the investigators express CD169 in 293T cells or other cells that do not express ACE2 to examine if CD169 alone can mediate infection in these cells? This would be preferable to do in a cell line that is highly sensitive to infection and not macrophages.

2. Have the authors validated the role of CD169 in modulating infection in vivo? Do CD169 KO mice have differences in SARS-CoV-2-mediated lung pathology? Conversely, and getting at the same question, can inhibition of CD169 via 7D2 in mice dampen SARS-CoV-2 mediated inflammation in infected mice?

3. How were the live SARS-CoV-2 virus infections done in the THP1/ACE2 and CD169-expressing cells? And in MDM expressing ACE2? Is there a wash step to remove unbound virus before quantifying infectious viral titers? Why is only the ACE2-expressing MDM mediating virus infection? Is there a possibility that SARS-CoV-2 live virus remains attached to ACE2 but not the CD169-expressing MDMs (Figure 3F) and this is driving the replication observed? This should be clarified.

Reviewer #3: Comments:

1. The findings in Figure 1 should be substantiated with actual SARS-CoV-2 infection. Latter studies were conducted with live virus infections using modified THP1 cells, but not primary MDMs. The studies in Figure 3 primarily featured infection data in primary MDMs transduced to over-express CD169 or ACE2, but not unmodified MDMs.

2. Is live virus infection of macrophages require TMPRSS2?

3. What are the sialylated motifs recognized by CD169 on SARS-CoV-2 spike protein? The findings in Figure 2 suggest that this is the case, but not conclusive data were presented.

4. How does the over-expression of CD169 compare to the physiologically relevant expression of CD169 on lung-resident macrophages (alveolar; interstitial or MDMs)?

5. The authors should include UV-inactivated virus to show the robustness of the RNA FISH staining in Figure 4A. Further, how do these findings compare with unmodified\\untransduced primary MDMs infected with SARS-CoV-2?

6. How are the authors able to detect negative-sense SARS-CoV-2 RNA? No evidence is provided by the authors beyond a citation that negative strand vRNA is detected in their assay.

7. Figure 5- The authors should include UV-inactivated controls and dose titration studies to demonstrate that the inflammation observed in primary MDMs is indeed due to virus infection and not an artifact in the media within the virus inoculum.

8. The authors have not adequately demonstrated that innate immune activation requires de novo viral RNA synthesis.

9. What is the in vivo role of MAVS in macrophages during SARS-CoV-2 infection?

**Part III – Minor Issues: Editorial and Data Presentation Modifications**

Reviewer #1: 1. The authors may want to consider showing that the abortive infection they observe is FcgR-independent by using blocking antibodies.

2. The induction of RLR-dependent inflammasome activation (IL-18 and IL-b) is interesting. Any ideas how these pathways are linked ?

3. Figure 5A and G: The induction of IL-1 beta and IL-18 in THP-1 CD169 but not in ACE2 expressing cells is also interesting and reveals activation CD169-entry dependent activation of inflammasomes. Also I am a little surprised by the expression of proinflammatory cytokines and IFNlambda despite remdesivir treatment in MDMs (Figure 5G). Although significantly lower than without RDV, expression levels are still 10-30 fold above the background suggesting a virus replication-independent sensing.

5. Figure 4C: With regard to point 4 and CD169-dependent inflammasome activation, are the THP-1/CD169 cells dying (pyroptosis) resulting in reduced N expression ? Were the expression measured wrt to a house keeping gene ? This would be required to interpret the data.

Reviewer #2: 4. How about different SARS-CoV-2 variant spikes? Are pseudotypes expressing delta or omicron spikes more infectious compared to the Wuhan-1 spike via the CD169 entry mechanism? Especially given that omicron may prefer the endocytic pathway, it would be particularly interesting to evaluate pseudotyped spikes expressing B.A.1 or other omicron sublineages.

5. Can the authors elaborate on the functional relevance of CD169 expression macrophages from COVID-infected people? While their CD169 data in THP1 cells is interesting, what does this mean in the broader context?

6. In the macrophage phagocytosis experiments that examined pseudotypes expressing SARS-CoV-2 spike, did the authors also examine a negative control NL63 spike to examine the background infectivity of these cells via the endocytic pathway. This control should also be shown in Figure 1. Currently, it is unclear what the background infection is with this system.

Reviewer #3: (No Response)

PLOS authors have the option to publish the peer review history of their article (what does this mean?). If published, this will include your full peer review and any attached files.

Reviewer #1: No

Reviewer #2: No

Reviewer #3: No
---

## [Decision Letter · Decision Letter 1]

6 Oct 2022

Dear Prof. Gummuluru,

We are pleased to inform you that your manuscript 'CD169-mediated restrictive SARS-CoV-2 infection of macrophages induces pro-inflammatory responses' has been provisionally accepted for publication in PLOS Pathogens.

Best regards,

Craig B Wilen, MD, PhD

Reviews Editor

PLOS Pathogens

Michael Diamond

Section Editor

PLOS Pathogens

Kasturi Haldar

Editor-in-Chief

PLOS Pathogens

orcid.org/0000-0001-5065-158X

Michael Malim

Editor-in-Chief

PLOS Pathogens

orcid.org/0000-0002-7699-2064

Reviewer Comments (if any, and for reference):

Reviewer's Responses to Questions

**Part I - Summary**

Reviewer #2: The authors have addressed this reviewer's comments/concerns. The revised version is greatly improved. I think this is interesting work and contributes to our understanding of SARS-CoV-2 pathogenesis. I recommend acceptance for publication.

Reviewer #3: The authors have adequately addressed the previous reviewers concerns. No additional concerns were noted.

**Part II – Major Issues: Key Experiments Required for Acceptance**

Reviewer #2: none

Reviewer #3: (No Response)

**Part III – Minor Issues: Editorial and Data Presentation Modifications**

Reviewer #2: none

Reviewer #3: (No Response)

PLOS authors have the option to publish the peer review history of their article (what does this mean?). If published, this will include your full peer review and any attached files.

Reviewer #2: No

Reviewer #3: No

---

## [Editor Report · Acceptance letter]

18 Oct 2022

Dear Prof. Gummuluru,

We are delighted to inform you that your manuscript, "CD169-mediated restrictive SARS-CoV-2 infection of macrophages induces pro-inflammatory responses," has been formally accepted for publication in PLOS Pathogens.

Best regards,

Kasturi Haldar

Editor-in-Chief

PLOS Pathogens

orcid.org/0000-0001-5065-158X

Michael Malim

Editor-in-Chief

PLOS Pathogens

orcid.org/0000-0002-7699-2064